# Microvascular immunity is organ-specific and remodeled after kidney injury in mice

Rebecca Rixen[1,6], Paula Schütz[1,6], Carolin Walter[2,6], Birte Hüchtmann[1], Veerle Van Marck[3], Barbara Heitplatz[3], Julian Varghese [2], Georg Varga[4], Dirk Foell[4], Thomas Pap [5], Hermann Pavenstädt[1] & Konrad Buscher [1]✉

Many studies analyze tissue-resident or blood-borne leukocytes to monitor disease progression. We hypothesized that the microvasculature serves as a distinct site for immune cell activity. Here, we investigate microvascular leukocyte phenotypes before, during and after acute kidney injury (AKI) in mice, uncovering unique characteristics in the kidney, liver, and lung. Using single-cell sequencing, we identify several immune cells that were up to 100-fold expanded in the kidney vasculature, including macrophages, dendritic cells (DC), and B cells. Regeneration after AKI is characterized by sustained remodeling of the renal microvascular interface. Homeostatic microvascular C1q[+] macrophages withdraw from the vascular barrier which is subsequently repopulated by new subsets, including CD11c[+]F480[+] and CD11c[+]F480[−] cells. These newly arrived macrophages exhibit enhanced phagocytic activity toward circulating bacteria and secretion of tumor necrosis factor, pointing to maladaptive repair mechanisms after AKI. These data suggest organ- and disease-specific microvascular immune dynamics which are not detectable through conventional blood and tissue analysis.

A comprehensive understanding of an organism's immune state is of profound clinical importance[1–3]. Venous blood samples are routinely employed to explore cell, protein and molecule concentrations. However, immunity is significantly shaped in lymphoid organs and local tissues. From an immunological perspective, the blood circulation primarily functions as a conduit, facilitating the transport of immune cells, mediators and metabolites to nearly every part of the body. Therefore, the extent to which blood biomarkers can capture the immune dynamics in health and disease remains an open question[4,5].

Organs harbor resident leukocytes that play a pivotal role in immunity. They encompass a spectrum of cells critical in innate and adaptive immune responses, including the mononuclear phagocyte system (such as monocytes, macrophages, and dendritic cells (DC)), natural killer (NK) cells, lymphocytes, and innate lymphoid cells (ILCs)[6]. While some of these cell types remain sessile within the tissue,

others engage in continuous recirculation through the lymphatic and circulatory systems[6]. The microcirculation acts as an interface between the tissues and the bloodstream, facilitating the extravasation of blood-borne cells into the parenchyma during homeostasis and inflammation. Beyond their transit function, endothelial cells also enable intravascular immune surveillance[7]. Some immune cells have been identified as preferentially residing within the vasculature. Non-classical monocytes and lymphocyte subsets, for instance, actively "patrol" the endothelium to detect cues of injury[8,9]. In atherosclerosis, these monocytes accumulate even in high-shear arteries to safeguard the endothelial barrier[10]. Kupffer cells, located intravascularly within the liver sinusoids, screen for blood-borne antigens and bacteria. In the lung, intravascular macrophages, marginating B cells, and NKT cells have been documented[11–13]. Hence, the microcirculation represents a topological niche of high significance for our understanding of

[1]Department of Medicine D, Division of General Internal Medicine, Nephrology and Rheumatology, University Hospital Münster, Münster, Germany. [2]Institute of Medical Informatics, University of Münster, Münster, Germany. [3]Institute of Pathology, University Hospital Münster, Münster, Germany. [4]Department of Pediatric Rheumatology and Immunology, University Hospital Münster, Münster, Germany. [5]Institute of Musculoskeletal Medicine, University Hospital Münster, Münster, Germany. [6]These authors contributed equally: Rebecca Rixen, Paula Schütz, Carolin Walter. ✉e-mail: konrad.buscher@ukmuenster.de

immunity. Nevertheless, it frequently eludes direct scrutiny, and microvascular immune cells at the blood-tissue interface have been rarely studied.

Here, we hypothesize that microvascular immune responses play a critical role in kidney injury and repair. Therefore, we implement a protocol to label microvascular leukocytes in different mouse organs and comprehensively analyze their numbers, phenotypes and inflammatory changes in direct comparison to matched peripheral blood and tissue samples.

## Results

We adopted an intravascular leukocyte staining protocol with intravenous injection of an anti-CD45.2 antibody prior to euthanasia[14]. A circulation time of 5 min did not lead to extravascular staining in the kidney, liver, and lung as confirmed by using the epithelial (= extravascular) markers EpCAM and E-cadherin (supplementary Fig. 1). Ex vivo staining of CD45 allowed to separate microvascular (CD45+ CD45.2+) and extravascular (CD45+ CD45.2−) leukocytes (Fig. 1a and supplementary Fig. 2). The spleen is a notable exception as the

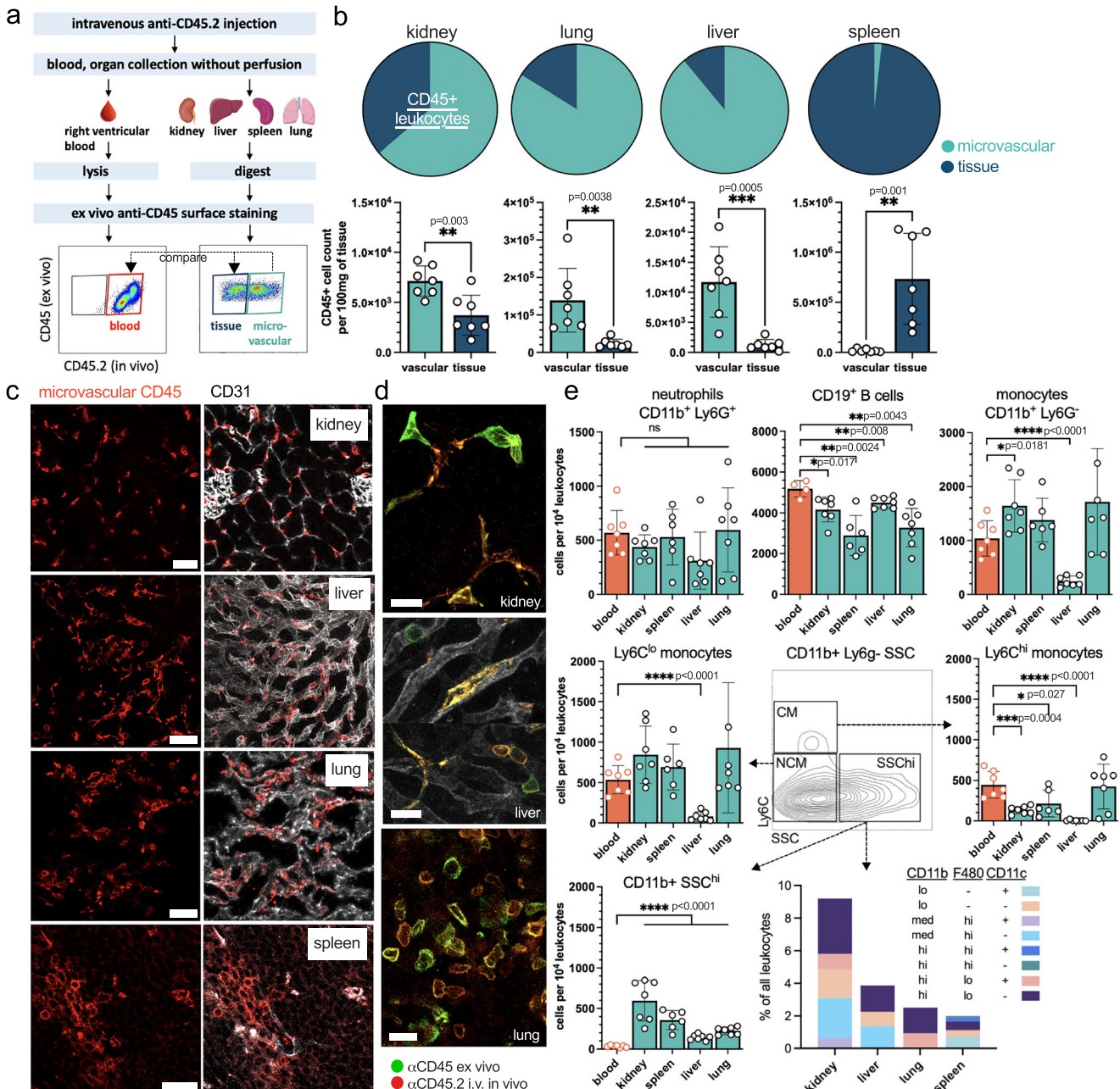

Fig. 1 | Microvascular leukocytes are abundant, organ-specific and differ from peripheral blood. a Experimental setup to detect microvascular leukocytes in mice, and compare them to tissue-resident and blood counterparts. Icons from the Servier Medical Art collection under CC BY 4.0. b The relation of microvascular and tissue-resident leukocytes in different healthy mouse organs. Ratios are shown in pie charts and absolute numbers as bar diagrams. Paired two sided t test. n = 7 independent experiments. Mean and SD are shown. c Visualization of microvascular leukocytes in healthy organs after intravenous injection of an anti-CD45 antibody. Scale bar 100 μm. d The combination of in vivo and ex vivo anti-CD45 staining highlights vascular and tissue-resident leukocytes. Intraluminal cell extensions (double positive) of perivascular cells (green) are predominantly found in the kidney. Scale bar 20 μm. Images are representative of 3 independent experiments. e Flow cytometry analysis of different intravascular leukocyte subsets in comparison to peripheral blood (red bar). Classical (CM) and nonclassical monocytes (NCM) are side-scatter (SSC) high and low, respectively. CD11b+ Ly6G− SSChi cells were further analyzed using F480 and CD11c. Paired two sided t test of blood versus each organ. Mean and SD are shown. Each dot represents one independent experiment. Source data are provided as a Source Data file.

microcirculation features open sinusoids with a discontinous endothelium. We did not perfuse the organs in situ to avoid losing marginating (temporarily adhered) cells in the microcirculation and to prevent possible disruption of the organ architecture[14]. Our main strategy was to compare the blood leukocyte composition in the renal microvasculture (CD45+CD45.2+ cells in the kidney cell suspension) with a time-matched sample of peripheral blood drawn from the right heart.

In the kidney, lung and liver, vascular leukocytes largely outnumbered parenchymal leukocytes to varying degrees (Fig. 1b). The high abundance of microvascular leukocytes was also evident in immunofluorescence stainings (Fig. 1c and supplementary Fig. 3). The combination of in vivo and ex vivo CD45 staining highlighted that few leukocytes are located extravascularly with intraluminal extensions, predominantly in the kidney (Fig. 1d). We assessed different leukocyte lineages in these organs by flow cytometry (gating strategy in supplementary Fig. 2). Microvascular cell populations were then compared to the peripheral blood sample in order to identify microcirculation-specific differences in leukocyte numbers and phenotypes. There was no difference in neutrophil numbers (Lin⁻ CD11b⁺ Ly6G⁺) between the blood and any organ vasculature (Fig. 1e). In contrast, B cells (Lin⁻ CD19⁺) were reduced in the microcirculation of most organs compared to blood (supplementary Fig. 2). Monocyte numbers (Lin⁻ CD11b⁺ Ly6G⁻ side scatter (SSC)ˡᵒ) significantly differed. Kidney, spleen, and lung showed a reduction of classical monocytes (Ly6Cʰⁱ SSCˡᵒ) that was not evident in nonclassical monocytes (Ly6Cˡᵒ SSCˡᵒ). A notable exception was the liver vasculature where monocytes are largely depleted (Fig. 1e). CD11b⁺ Ly6G⁻ myeloid cells include a SSCʰⁱ subset that indicates a higher granularity (Fig. 1e). They were increased in the vasculature of all organs, predominantly in the kidney, and were not detectable in the blood. Further analysis using the markers CD11b, F480 and CD11c revealed an organ-specific composition of cell subsets with a macrophage or dendritic cell signature (Fig. 1e, gating in supplementary Fig. 2).

To detect unsupervised phenotypes and transcriptional states of microvascular leukocytes we performed single cell RNA sequencing. Leukocytes in the renal tissue, the renal microvasculature (MV) and in the blood of one mouse were sorted, multiplexed and processed together on the same microwell device for library preparation. Using 4 healthy mice, 33.474 leukocytes were sequenced (13.828, 13.408 and 6.238 in blood, microcirculation and tissue, respectively), resulting 27 CD45⁺ leukocyte clusters (Fig. 2a, supplementary Fig. 4 and supplementary Data 1 and 2). The results were confirmed in a downsampled data set with an equal cell distribution (1:1:1) for each compartment.

Demultiplexing allows for assigning each cell to its origin (renal tissue vs. renal microvasculature vs. blood), demonstrating a cell type and compartment-specific distribution (Fig. 2b–e, supplementary Fig. 4, and supplementary Data 1). As expected, the populations of innate lymphoid cells (ILC) types 1-3 (clusters 6, 12, 18, 21) had a strong tissue signature (Fig. 2d, e and supplementary Fig. 4). Interestingly, a number of leukocytes were preferentially found in peripheral blood mostly sparing the renal circulation (clusters 1, 3, 4, 16, 20, Fig. 2c–e). This included lymphocytes, confirming our flow cytometry data. Vice versa, seven cell types populated the renal microcirculation to a significantly higher amount compared to the blood (Fig. 2c, d). This included cluster 0 as highly abundant C1Q⁺ ADGRE1⁺ ITGAX⁺ macrophages. They were almost 100-fold enriched in the renal vasculature compared to blood but not significantly enriched compared to the tissue (Fig. 2c–e), suggesting a localization at the blood-kidney interface. Six other leukocytes populations were also enriched in the renal microvasculature (5- to 50-fold enriched, cluster 2, 7, 15, 17, 18, 25) including XCL1⁺ C1Q⁺ macrophages, plasmacytoid DC, conventional DC type 1, IL2, and a B cell subset (Fig. 2c, d and supplementary Fig. 4).

Cluster 2 was exclusively present in the renal vasculature but absent in the blood and kidney tissue (Fig. 2c–e and supplementary Data 1), pointing to a specialized endovascular B cell subset.

Reclustering of all B cells showed a gradient between the blood and microcirculation (Fig. 3a, b). Of all renal B cells, 86% were microvascular and 14% were tissue-resident. Subclusters 6, 4 and 7 were 5-, 12- and 15-fold increased in the renal vasculature compared to blood, respectively (Fig. 3c). Clusters 0 and 5 were significantly upregulated in the blood, and cluster 1, 4, 6 and 7 were enriched in the renal microvasculature (Fig. 3d). Slingshot, a lineage inference tool, identified two developmental trajectories from the main blood-borne B cell cluster (cluster 0) to the microvascular clusters (Fig. 3e). We found the transcription factor Early Growth Response-1 (EGR1) and Interferon Regulatory Factor 4 (IRF4) as main driver genes underlying this phenotype switch in our data set. The transcriptional shift included differentially regulated gene pathways in microvascular B cells including Fc receptor signaling (Fig. 3f). Using immunofluorescence imaging with intravascular staining of CD19, vascular B cells were found almost exclusively in glomerular capillaries whereas tubular and medullary regions were largely depleted (Fig. 3g). These data suggest an endovascular B cell phenotype switch elicited by glomerular capillaries.

Next, we investigated how disease affects microvascular leukocytes. We used two different mouse models of inflammation: acute inflammation by semi-sterile peritonitis (INF) and acute kidney injury (AKI) by ischemia reperfusion (Fig. 4a). INF represents a sham control which allows to distinguish peritonitis- from AKI-related effects. We hypothesized that AKI remission leads to long-lasting alterations of the microvascular niche. Therefore, next to the acute phase (day 1 post surgery) we also included a late time point (day 12 post surgery, INF-reg and AKI-reg, Fig. 4a). The kidney function was assessed by serum creatinine and renal histology, confirming injury at AKI day 1 and renal reconstitution at day 12 (supplementary Fig. 5). In these disease models we could confirm that the injected anti-CD45 antibody does not leak into the tissue (supplementary Fig. 1).

We found that the total number and composition of microvascular leukocytes changed in function of time, organ and disease model, often being independent of measurements in the peripheral blood (Fig. 4b–d). During the acute phase of AKI, leukocytes increased about 16-fold in the splenic circulation compared to the blood (Fig. 4c). AKI-reg was associated with a 20- and 7-fold leukocyte increase in the liver and kidney tissue, respectively (Fig. 4c). In the cell subset analysis (Fig. 4d), the absolute numbers were normalized to the untreated blood sample to emphasize changes in relation to blood (absolute numbers are shown in supplementary Fig. 6). Ly6G⁺ neutrophils increased 4- to 8-fold in the blood during the acute phase of AKI and INF, and returned almost to baseline after recovery (Fig. 4d). However, this pattern did not reflect the neutrophil presence in the microcirculations of the organs: in the INF model, the blood neutrophil increase was lower or absent in the kidney, liver and spleen microcirculation (Fig. 4d). In contrast to blood, neutrophils remained enriched in the lung and liver microcirculation after AKI recovery (Fig. 4d). Nonclassical monocytes and SSCʰⁱ monocytes were 16-fold and 64-fold upregulated in the liver microvasculature in AKI-reg, respectively (Fig. 4d). Lymphoid cell types were mostly downregulated in the microvasculatures to varying degrees compared to blood (supplementary Fig. 5). Together, these data suggest that inflammation affects microvascular leukocyte dynamics in an organ- and disease-specific manner that cannot be fully captured using peripheral blood analysis.

To study microvascular immune cells in kidney regeneration after injury, we applied single cell RNA sequencing of multiplexed leukocytes sorted from the blood, renal microcirculation and renal tissue in AKI-reg (Fig. 5a). The results were integrated with data from healthy kidneys. 30 leukocyte clusters could be detected based on 79.328 cells. Demultiplexing assigned the cells to the specific condition (UT, AKI-reg) and anatomical compartments (tissue, blood, microvasculature) (Fig. 5b, c, supplementary Fig. 7, supplementary Data 1 and 2). Compared to the healthy kidney, 7 leukocyte subtypes were significantly increased in AKI-reg (up to 14-fold in the renal vasculature and up to

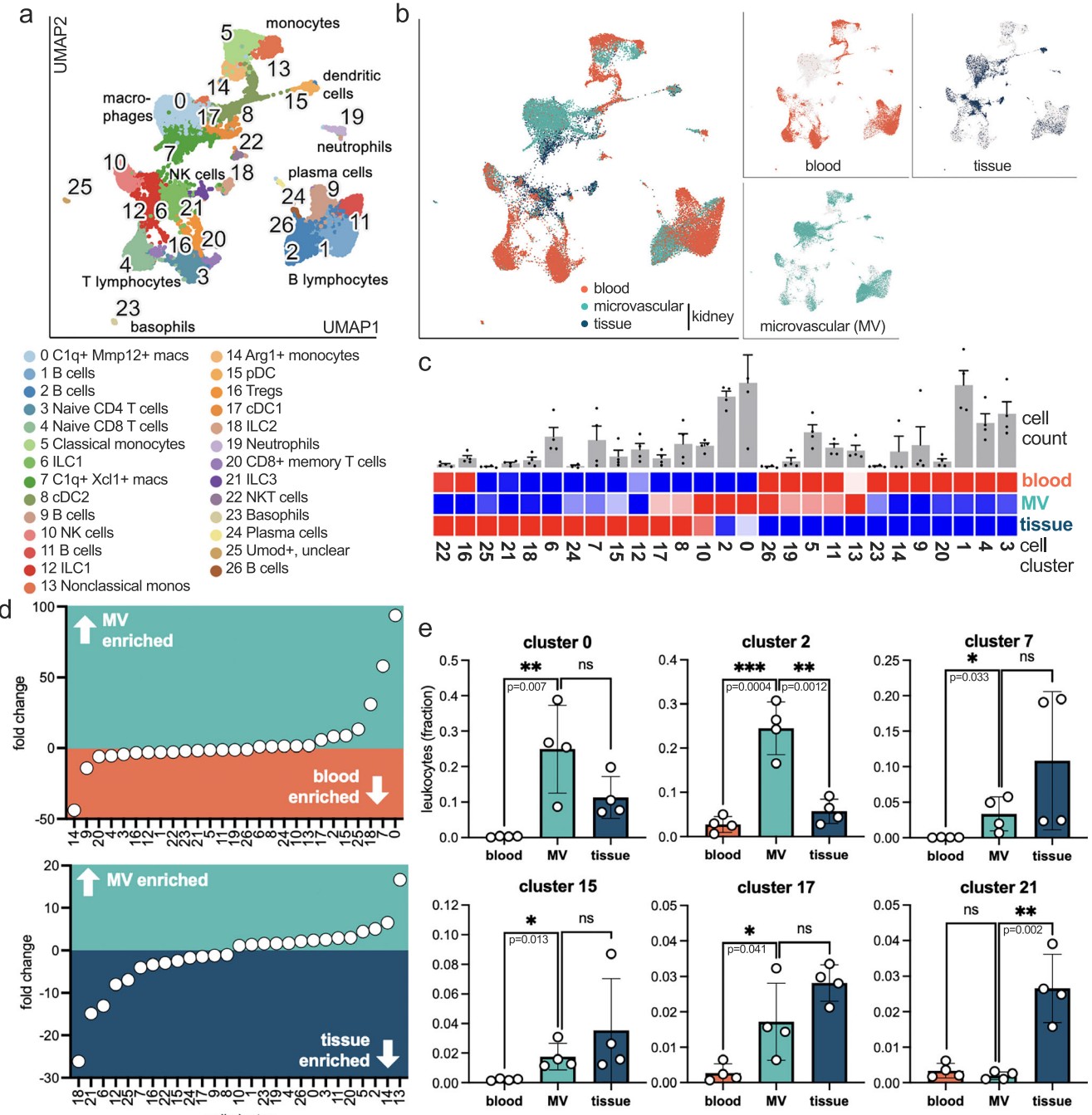

**Fig. 2 | Unsupervised phenotyping of microvascular leukocytes in the healthy mouse kidney. a**, **b** Single cell sequencing of leukocytes sorted from blood, the renal microvasculature (MV) and the renal tissue. UMAP visualization and annotation of 27 leukocyte subtypes. $n = 4$ independent experiments. **c** Abundance of cell cluster specific leukocytes in the three compartments. Blue is the lowest number and red the highest number for each column. The top bar graphs represent the total cell count of each cluster as mean ±+/− SD of $n = 4$ independent experiments.

**d** Ranked fold change of microvascular versus peripheral blood (top), and microvascular versus tissue-resident (bottom) leukocytes for each cell cluster. **e** Selected cell clusters are shown in the three compartments as fraction of all detected leukocytes. Unpaired two sided $t$ test between MV and blood, and MV and tissue. All clusters are shown in supplementary Fig. 4. Mean and SD are shown. Each dot represents one independent experiment. Source data are provided as a Source Data file.

42-fold in the tissue, none in blood; Fig. 5d). Of these, 4 cell types (non classical monocytes, classical monocytes, neutrophils and *CCL17*[+] *TNF*[+] cDC2 as clusters 11, 2, 6, 8, respectively) were detectable in the renal microcirculation (Fig. 5d, e). Three cell types were upregulated in both the tissue and the microcirculation, suggesting a localization at the blood-kidney interface (Fig. 5e). The endovascular B cell subset (cluster 2) was significantly reduced after AKI recovery. Interestingly, one neutrophil phenotype (cluster 6) was enriched in the tissue and the

microvasculature, while another (cluster 13) was only present in the tissue, suggesting transcriptomic changes in function of neutrophil trafficking. None of these alterations could be detected by peripheral blood analysis (Fig. 5d, e).

Macrophages and dendritic cells (DC) are major effectors in ischemic AKI[15]. To obtain more insights, the respective cell subsets (clusters 0, 8, 12, 16, 19; excluding monocyte subsets) were subclustered and reanalyzed, resulting in 8 cell clusters (Fig. 6a) with distinct gene

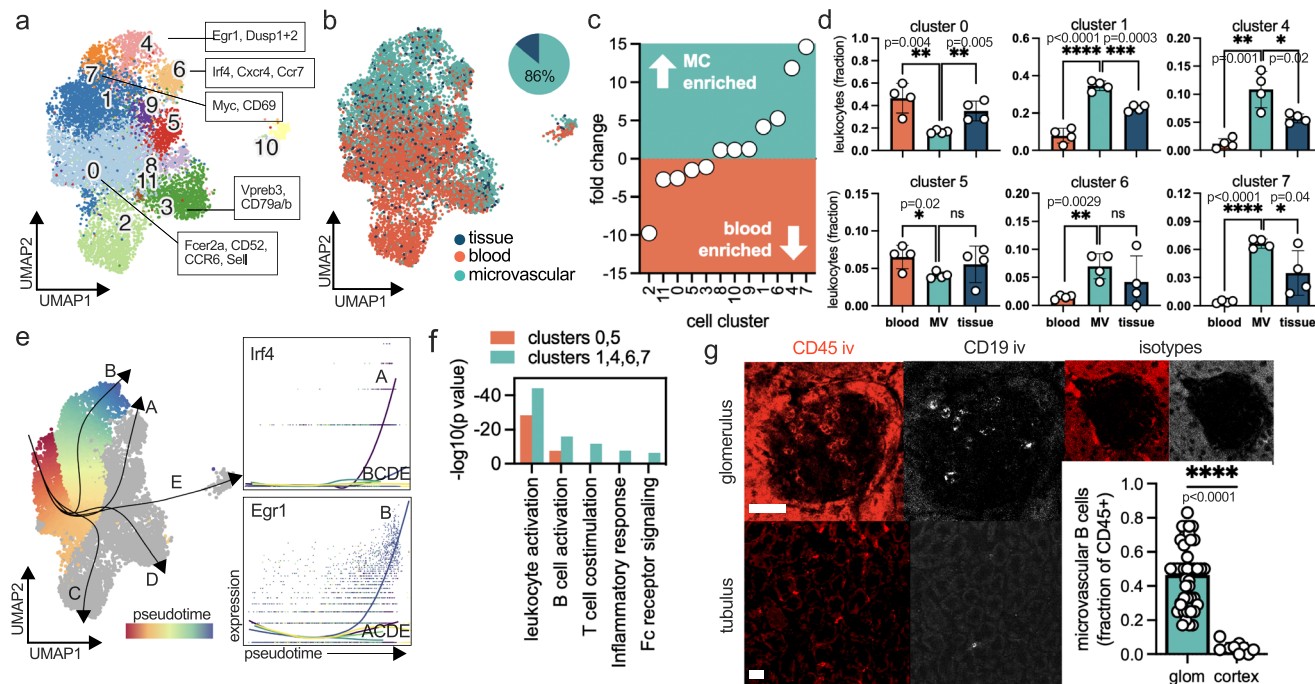

**Fig. 3 | A transcriptional shift of microvascular B cells in glomerular capillaries. a** Reclustering of the B cell clusters shown in Fig. 2. Selected differentially expressed (DE) genes are annotated. **b** Demultiplexing of all B cells into their origins (peripheral blood, tissue or microvasculature, MV). Of all B cells in the kidney, 86% are found in the microcirculation. **c** Ranked fold change of microvascular versus peripheral blood B cells. **d** Absolute B cell numbers in all clusters with significant changes. Paired two sided *t* test. Mean and SD are shown. Each dot represents one independent experiment. Source data are provided as a Source Data file. **e** Slingshot analysis shows transcriptional phenotype trajectories from the main B

cell cluster of the peripheral blood (cluster 0, significantly enriched in blood) to microvascular B cells of the kidney (clusters 7,4, and 6): unique upregulation of Irf4 and Egr1 in the microvascular B cell trajectories A and B, respectively. **f** Functional annotation of DE genes in peripheral blood B cells (cluster 0, 5; significantly enriched in blood) versus microvascular B cells of the kidney (clusters 1,4,6,7; significantly enriched in the microcirculation). **g** Intravascular anti-CD19 B cell staining in the healthy kidney identifies the glomerular capillaries as the main B cell residence. Scale bar 50 μm. Each dot represents one microscopy image, n = 9 (tubulus) and *n* = 44 glomerulus, from *n* = 3 biological replicates. Mean and SD are shown.

expression patterns (Fig. 6b and supplementary Data 2). Demultiplexing allowed to visualize changes in cell phenotypes between untreated and AKI-reg conditions in the microvascular and tissue compartment (Fig. 6c). In the healthy mouse kidney, 67% of all macrophages and DC were in contact with the microcirculation. In AKI-reg, the ratio was reduced to 26% (Fig. 6d). We found that recovery after AKI is characterized by a breakdown of the microvascular interface. The largest microvascular population of *C1Q*+ macrophages (cluster 0) was about 3-fold reduced in the microvasculature after AKI recovery but remained high in the tissue, indicating an exclusion from the endoluminal space (Fig. 6e, f). The top differentially expressed genes include *MMP12, CXCL1, C1Q, COL14A1* and *TGFB3* (supplementary Data 2). A total of four (clusters 1, 2, 4, 5; supplementary fig. 8) new macrophage and DC subsets significantly enrich the microvascular interface after AKI (Fig. 6e), including *CCR2+ CD300A+ CD163+* monocyte-derived, M2-polarized macrophages (cluster 2; 5-fold increase; Fig. 6f).

We next aimed to identify specific functions of microvascular cells. As validated surface markers for the identified macrophage/DC cell subsets are lacking, we used the common markers CD11c and F480 (genes *ADGRE1* and *ITGAX*, respectively). Double positive staining is indicative for cell clusters 0, 1, 2, 4 and 5, whereas Cd11c single positive expression can be found in cluster 3 (Fig. 6b). Using immunofluorescence microscopy, the increase of CD11c+ cells in the kidney microvasculature after injury could be confirmed (Fig. 6g). Flow cytometry analysis revealed that the microvascular CD11c+ cell population includes the subsets CD11c+F480+ and CD11c+F480− (Fig. 6h). As cells of the microvascular interface are in contact with circulating pathogens, we tested the capacity to phagocytose bacteria in the bloodstream (Fig. 7a). Twenty minutes after iv injection of pH-sensitive E.coli bioparticles, the phagocytotic activity of microvascular CD11c+

cells was low in healthy kidneys, and doubled after recovery from AKI (Fig. 7b, c). Phagocytosis could only be detected in microvascular CD11b+CD11c+F480+ macrophages but not in CD11b+CD11c+F480− dendritic cells (Fig. 7c). In both cell types, CD206 was highly upregulated after AKI recovery, which could not be detected in corresponding cells residing in the tissue (Fig. 7d). Tumor necrosis factor (TNF) is a central cytokine that is produced by macrophages and DC, and is known to exacerbate kidney injury[16,17]. Using intracellular cytokine analysis by flow cytometry and ex vivo LPS stimulation, we found that TNF secretion is significantly increased predominantly in microvascular and less in tissue-assigned CD11c+F480+ cells from healthy kidneys (Fig. 7e). In contrast, microvascular CD11c+F480− dendritic cells hardly secrete TNF in homeostatic conditions but significantly increase their capacity to secrete TNF after kidney injury. This effect is not detectable in tissue-resident CD11c+ counterparts (Fig. 7e).

## Discussion

The composition of immune cells in large veins differs significantly from the microcirculation, and these differences vary between organs. In the healthy kidney, seven immune cell types were up to 100 times more abundant in the microvasculature compared to peripheral blood, where these cells were rare or absent. In two disease models, we observed organ-specific changes in microvascular leukocytes that were not reflected in the peripheral blood or the tissue. These results highlight a key limitation of standard blood tests, as they fail to capture the unique immune activity occurring in the microcirculation.

The microvascular barrier comprises various cellular components, including endothelial cells, the basement membrane, and pericytes[7,18]. Our investigation adds a further layer of structural complexity to this established concept: a heterogeneous assembly of

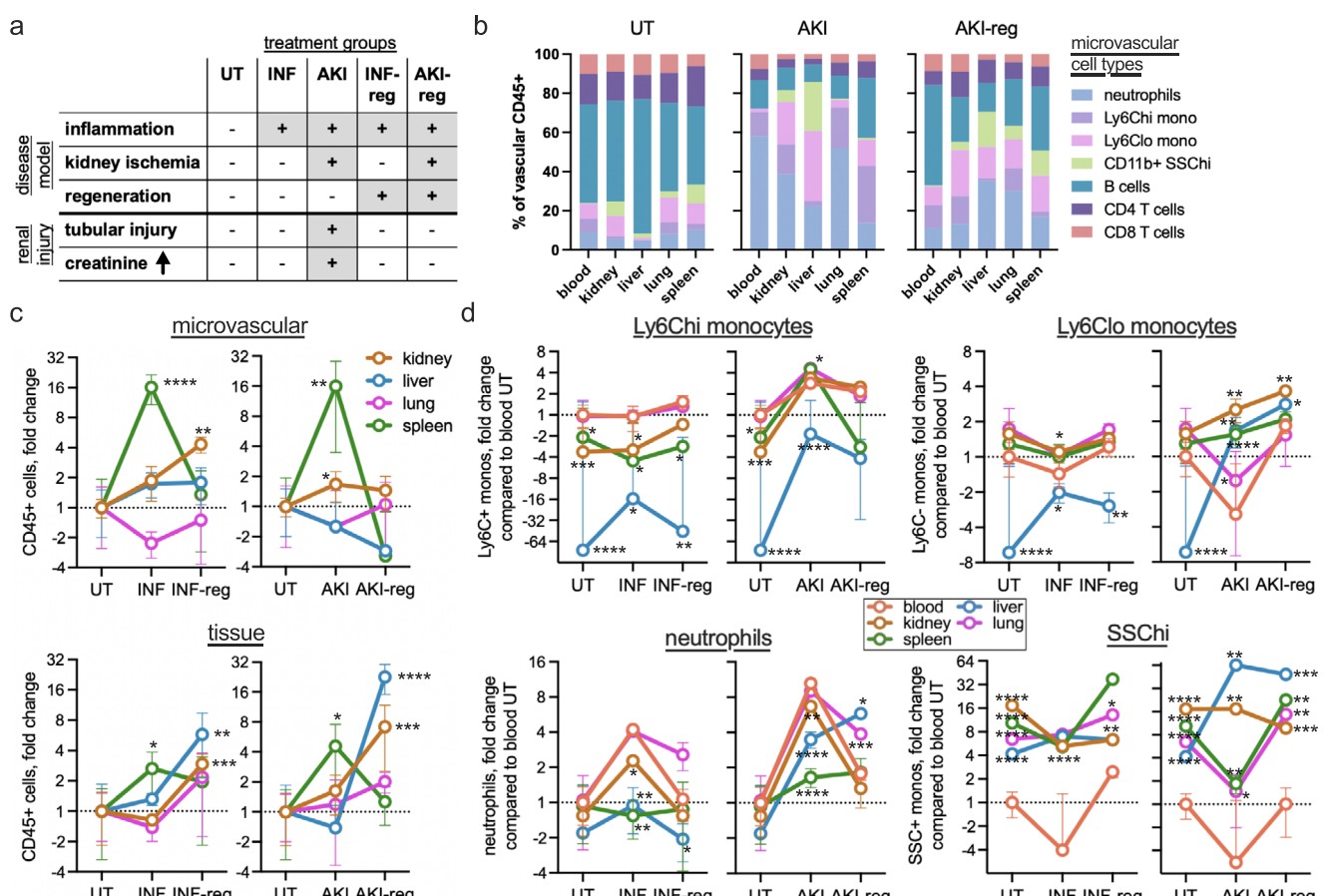

**Fig. 4 | Disease-specific alterations of the microvascular immune landscape across different organs. a** Overview of the disease conditions. INF = inflammation by semi-sterile peritonitis. AKI = acute kidney injury. INF-reg/AKI-reg = full regeneration 12 days after surgery. UT = untreated. **b** Microvascular leukocyte subtypes in AKI and AKI-reg. Mean and SD are shown. $n = 7$ (UT), 3 (INF, INF-reg), 7 (AKI), 4 (AKI-reg). **c** The total number of CD45+ leukocytes in selected organs expressed as relative to untreated (UT, equals 1). One way Anova with Dunett's multiple comparison showing significant changes compared to UT in each organ. Mean and SD are shown. **d** Microvascular neutrophils, monocytes and CD11b+ Ly6g− SShi cells in different organs and the peripheral blood expressed as relative to blood untreated (equals 1). Unpaired two sided $t$ test showing significant changes comparing the organ microvasculature to the blood. $n = 3$–7, details are provided in the source data sheet. Mean and SD are shown. $*p < 0.05$, $**p < 0.01$, $***p < 0.001$, $****p < 0.0001$.

organ- and vessel-specific immune cells that are integrated into the microvascular barrier with contact to circulating molecules in the bloodstream. In the healthy kidney, this includes B cells and different subsets of dendritic cells and macrophages. The cells can reside strictly in the vessel lumen or occupy the perivascular space with intraluminal extensions[19,20]. Another possibility is a perivascular position with exposition to blood borne molecules via transendothelial transport mechanisms[21,22]. We found endovascular B cells confined to glomerular capillaries, pointing to specific interactions with the highly specialized glomerular endothelium[23]. These cells displayed a distinct vascular phenotype with a transcriptional upregulation of the maturation-associated factors *IRF4* and *EGR1*[24,25]. Also T lymphocytes have been described to enrich in glomerular capillaries where they can interact with nonclassical monocytes during glomerulonephritis[9]. Intravascular naive B cells have been previously described in the heart microcirculation[26]. Adoptive transfer of circulating B cells replenished the microvascular B cell pool of the heart in B-cell deficient mice[26], suggesting recruitment from systemic reservoirs. Macrophages and DC constituted the largest microvascular leukocyte population in the kidney, including conventional DC1 (*IRF8+ FLT3+ CST3+*), plasmacytoid DC (*IRF8+ IRF7+ TLR7+*) and *C1Q+ ADGRE1+ ITGAX+* macrophages (*MMP12+ CSF1R+*, and *XCL1+ CXCL10+*). These populations have been partially visualized in CX3CR1-GFP/CD11c-YFP reporter mice that feature three perivascular cell populations with vascular extensions[19,27].

ILC2 were predominantly detectable in the tissue with a small enrichment in the microvasculare. A previous report detected IL5-producing ILC2s in the perivascular space of renal arteries[28]. Further experiments are requried to elucidate the location and function of these microavscular cell types.

Inflammation leads to vascular permeability and leukocyte extravasation, making the microcirculation a focal point of injury in several diseases[29]. The presence of immune cells at the microvascular barrier sensing molecules in the bloodstream carries significant functional implications. Immune complexes, antibodies, danger-associated molecular patterns (DAMPs)[30] or other mediators in the plasma could trigger immediate cell activation, circumventing other regulatory components within the microcirculation. Endovascular leukocyte dynamics have been recently highlighted in chronic fibrosis of the liver, where luminal monocytes aggregate and substitute Kupffer cell functions[31]. Our data indicate that the microvascular interface of the regenerated kidney exhibits sustained damage over an extended period of time. Here, a structural breakdown of the microvascular barrier persists, with most *C1Q+* macrophages losing contact with the bloodstream, becoming concealed within the tissue and expressing inflammation- (*MMP12*[32]) and kidney fibrosis-related genes (*COL14A1* and *TGFB3*[33]). The mechanism remains to be identified and could include cell migration, retraction of endoluminal extensions or structural changes of the vascular barrier. In contrast, *CCL17+* DC and monocyte-

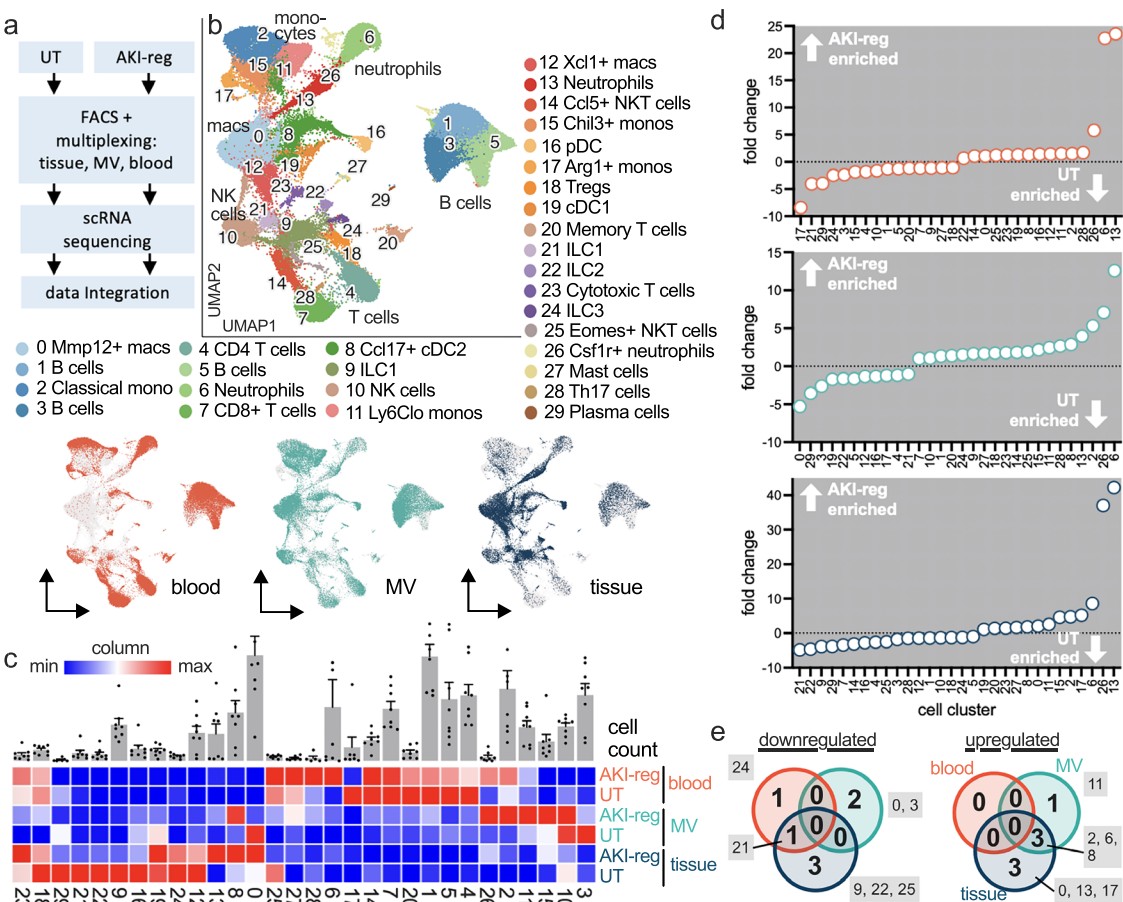

**Fig. 5 | Long-term remodeling of the microvascular leukocyte landscape after recovery of kidney injury. a** Experimental setup to integrate AKI-reg (*n* = 4) and untreated (UT, *n* = 4) single cell sequencing data of the kidney. MV = microvascular. **b** Leukocyte cell clustering, UMAP visualization, subtype annotation and demultiplexing. **c** Heatmap visualizing relative abundance of each cell cluster. Blue is the lowest number and red the highest number for each column. The top bar graphs represent the total cell count of each cluster as the mean ± SD of *n* = 8 independent experiments. **d** Ranked fold change of each cell cluster in the blood (top),

microvascular (center) and tissue (bottom) compartment. Negative and positive numbers show enrichment in the UT and AKI-reg condition, respectively. Cell cluster fractions are shown in supplementary Fig. 7. **e** Up- and downregulated cell clusters in AKI-reg compared to UT. Only cluster numbers with significant changes (unpaired two sided *t* test) are shown (gray boxes). Overlapping circles indicate that the same change can be detected in both (or all three) compartments. Source data are provided as a Source Data file.

derived macrophages enrich in the microvascular interface and display altered functions including phagocytosis and TNF secretion. TNF is a known driver of tubular injury and immune cell recruitment in AKI[17,34]. CD11c[+] cells were shown to be the main TNF-producing cell type in renal ischemia reperfusion, and in vivo depletion of DCs attenuated TNF secretion and tissue injury[17]. We therefore assume that the elevated TNF synthesis in CD11c[+]F480[-] DC is a sign of maladaptation after injury. Of note, the TNF secretion of microvascular cells was higher compared to tissue-resident counterparts, suggesting a major role of microvascular immunity in post-AKI dynamics. Together, the changes in microvascular immunity may contribute to the clinical observation that AKI elevates the overall risk of developing chronic kidney disease[35] and recurrent AKI[36].

There are limitations to be considered. Our experimental setup only provides a snapshot in time. Thus, we cannot extract temporal information such as disease dynamics, directional cell migration or intraluminal patrolling. Second, vessel density and intra-organ blood volumes were not accounted for. Literature reports relatively consistent values for kidney, lung, and liver in mice, suggesting that our findings are unlikely to be significantly skewed by these parameters[37,38]. Third, the spleen data need to be interpreted cautiously as the sinusoid architecture does not allow clear definitions of the intra- and extravascular space using our protocol. In general, we assume that

intravenous antibody application leads to direct labeling of leukocytes. It is possible, however, that in some tissues the antibody is rapidly (within a few minutes) shuffled via active transendothelial transport mechanisms (i.e. clathrin coated vesicles or endothelial caveolae) to adjacent leukocytes in perivascular locations[21,22]. We currently lack specific surface markers for the unique labeling of the identified cell subsets. Further work is required to resolve the biological heterogeneity of microvascular immune cells. Finally, our microvascular cell atlas does not provide topological data so that distinct entities of the vascular tree such as arterioles, capillaries or venules cannot be distinguished. Particularly in the complex kidney architecture anatomical considerations will be important to understand the precise biological roles of different microvascular leukocytes.

Collectively, our data highlight abundant, heterogeneous and organ-specific leukocytes that are located in the microvasculature and escape peripheral blood analysis. In the kidney, microvascular phagocytes are major players in repair mechanisms after AKI. We posit that microvascular immune dynamics are critical for a more comprehensive understanding of immunity and disease perturbations.

## Methods

### Antibodies and reagents

Antibodies and reagents are listed in supplementary Data 3.

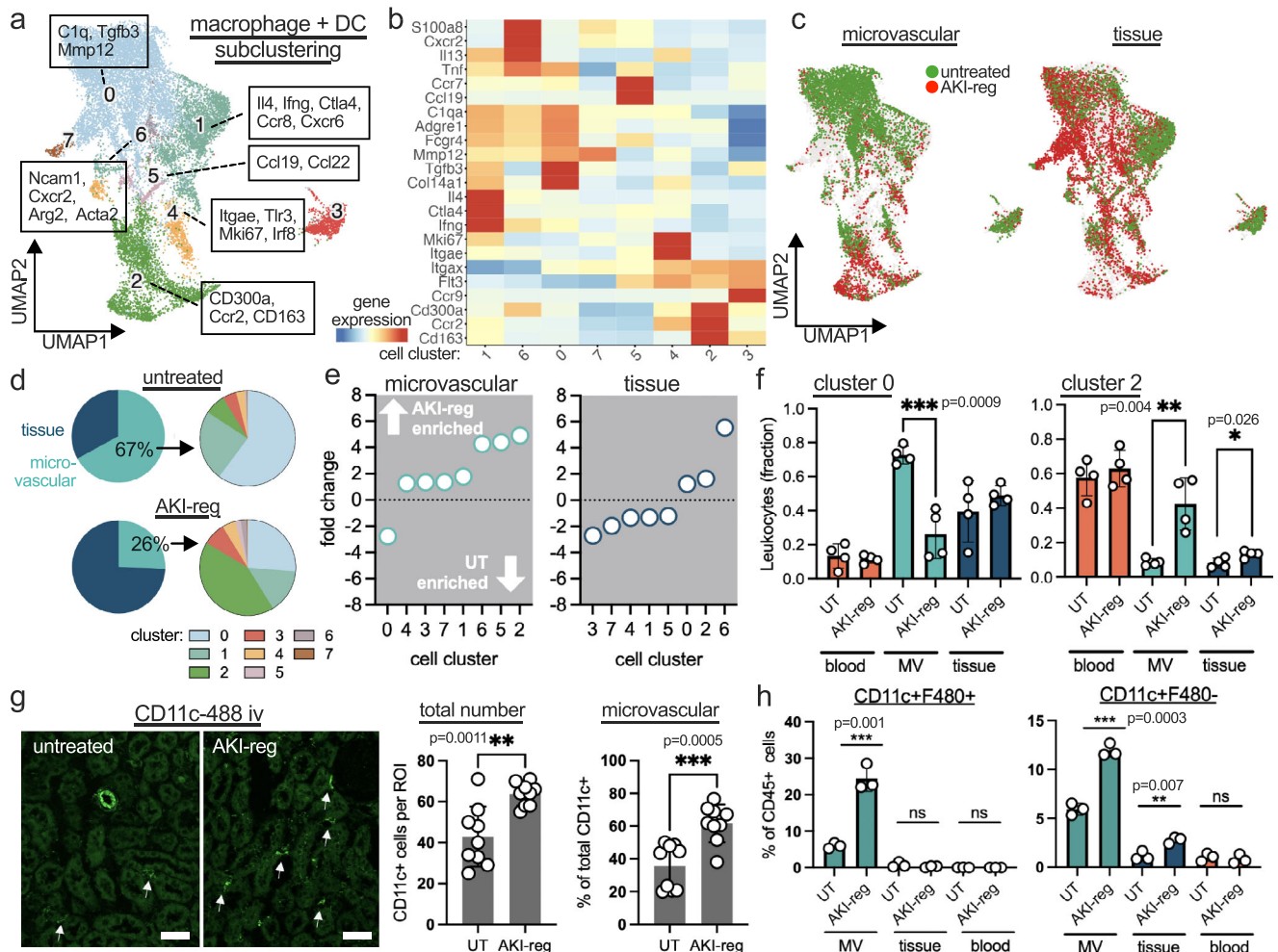

**Fig. 6 | Microvascular macrophage and DC dynamics after recovery of kidney injury. a** Reclustering and UMAP visualization of renal macrophages and dendritic cells (DC) in the data set shown in Fig. 5 (untreated and AKI-reg conditions). Differentially expressed genes are annotated. **b** Heatmap of selected genes with hierarchical clustering and relative expression. **c** Demultiplexing into microvascular (left) and tissue leukocytes (right). Cells detected in the untreated (UT) and AKI-reg conditions are green and red, respectively. **d** The ratios of microvascular (MV) and tissue-resident macrophages/DC are shown. A cluster subanalysis of the MV cell fraction is shown on the right. **e** Fold change analysis of each cell cluster. Positive numbers indicate enrichment in AKI-reg. **f** Detailed analysis of clusters 0 and 2 as the most up- and downregulated microvascular clusters. Unpaired two

sided *t* test. Mean and SD are shown. The number of dots represents independent biological replicates (*n* = 4). **g** Analysis of intravascular CD11c⁺ cells in UT and AKI-reg kidneys using immunofluorescence microscopy. The total (intra- and extravascular) number and microvascular cells were counted in 3 cortical regions of interests (ROI) from 3 independent biological replicates. Unpaired two sided *t* test. Mean and SD are shown. Scale bar 50 μm. **h** Flow cytometry analysis of CD11b⁺ Ly6G⁻ CD11c⁺ F480⁺/⁻ leukocytes in the renal microvasculature (MV), tissue and peripheral blood. Unpaired two sided *t* test. Mean and SD are shown. The number of dots represents independent biological replicates (*n* = 3). Source data are provided as a Source Data file.

## Mice and disease models

All animal experiments were approved by the authorities (Landesamt für Natur, Umwelt und Verbraucherschutz, 81-02.04.2019.A253) and performed in accordance with the animal protection guidelines of Germany. C57BL/6 mice were purchased by Charles River and bred in house. Only male mice were used because the kidney damage in the IRI model is gender specific[39,40]. The animals were housed at 21 °C and 40% humidity using a 12:12 light cycle. Deep anesthesia and cervical dislocation was used for euthanasia. Mouse models of peritonitis and bilateral renal ischemia/reperfusion injury (IRI) were studied at day 1 (INF/AKI) and day 12 (INF-reg/AKI-reg) post injury.

Buprenorphine (0.1 mg/kg) was administered subcutaneously to 8–20 week-old male mice 20 min prior to surgery. Mice were anesthetized with a 1,5–2% isoflurane/oxygen mixture, placed on a thermoregulated pad to maintain body temperature, and a midline abdominal incision was made. For AKI-IRI induction, both renal pedicles were carefully dissected and clamped with an atraumatic vascular clamp for 35 min. For inducing semi-sterile peritonitis (INF, INF-reg),

the cavity was left open for 35 min and the gut was mobilized similar to AKI-IRI. Animals received buprenorphine via the drinking water (0.009 mg/ml) up to 3 days post-surgery.

## Intravascular stainings

To study microvascular leukocytes, the vasculature was not flushed prior to organ collection[14]. Hence, blood-borne non-interacting leukocytes were present in the explanted organ. To account for this effect, we used peripheral blood as a reference. Of note, the fraction of microvascular (MV) leukocytes also contains blood borne leukocytes, and the comparison to peripheral blood allows to identify microcirculation-specific cell types. We opted for right ventricular blood as it closely resembles the composition of blood in larger peripheral veins typically accessed for venipuncture in human patients. For intravascular leukocyte stainings, a Phycoerythrin (PE)-labeled rat anti-mouse CD45.2 antibody (2 μg) was injected intravenously (supplementary Data 3). After 5 min of incubation, blood was collected by right cardiac puncture and organs were harvested without vascular

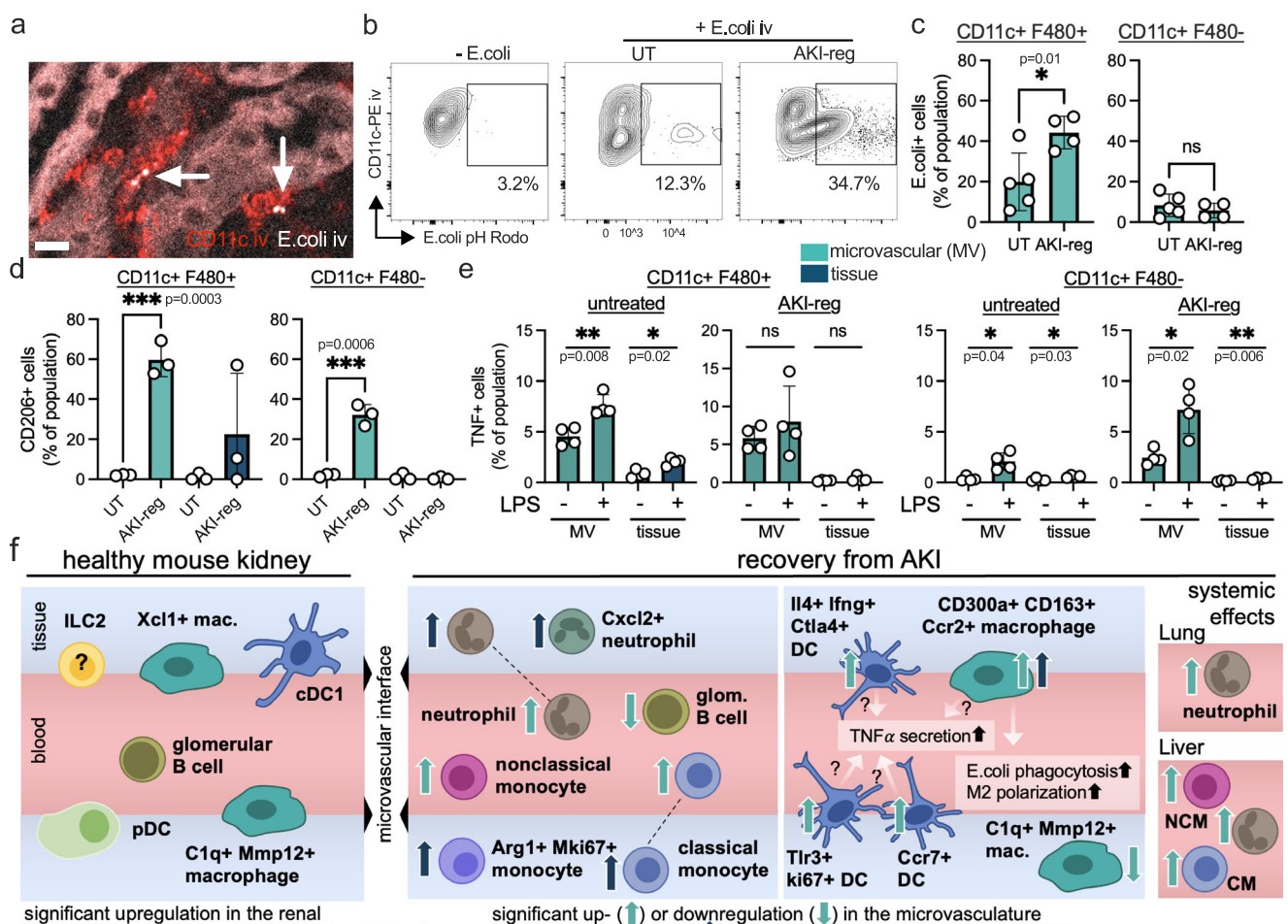

**Fig. 7 | Phagocytosis, M2 polarization and TNF secretion in microvascular macrophages and DC of the kidney. a** After iv-injection of pH-sensitive E.coli phagocytosis can be detected in renal intravascular CD11c⁺ cells within 5 min (arrows). Scale bar 10μm. **b**, **c** Flow cytometry analysis shows microvascular phagocytosis in healthy kidneys that increases significantly after recovery from kidney injury (AKI-reg). Unpaired two sided *t* test. Gating on microvascular CD45.2⁺CD11c⁺ leukocytes. Mean and SD are shown. **d** CD206 was analyzed in the microvascular (left bars) and tissue-resident (right bars) CD11c⁺ F480⁺ and CD11c⁺ F480⁻ leukocyte populations by flow cytometry. Unpaired two sided *t* test. Mean and SD are shown. **e** TNF secretion was determined by intracellular flow cytometry in microvascular and tissue resident CD11b⁺ CD11c⁺F480⁺ and CD11c⁺F480⁻ cell populations with and without ex vivo LPS stimulation. Mean and SD are shown. Paired two sided *t* test. The dots in all bar graphs represent independent biological replicates. Source data are provided as a Source Data file. **f** Main findings summarized as graphical abstract.

perfusion. Flow cytometry analysis allowed to gate for microvascular and tissue leukocytes. Isotype controls are shown in supplementary Fig. 2. To exclude the staining of the extravascular cells using this protocol, the epithelial cell antibodies anti-EpCam (CD326, APC-labeled) and anti-E-cadherin were injected (supplementary Fig. 1). Validation experiments were performed for untreated mice, AKI and AKI-reg (supplementary Fig. 1). For imaging experiments, 10 μg of fluorescently-labeled antibodies (CD45.2-AF594, CD31-AF647) were injected intravenously.

**Organ digestion and flow cytometry**
Heparinized blood was incubated with Fc Block (Biolegend) for 10 min on ice followed by antibody staining for 30 min. Erythrocytes (RBC) were lysed for 10 min at room temperature (RT) using 1x RBC Lysis/Fixation solution (Biolegend). Samples were washed with phosphate buffered saline (PBS) at 400 × *g* for 5 min and cell pellets were resuspended in PBS with 1% fetal calf serum (FCS). Organs were minced into small pieces, and kidney, liver, and lung were digested in 1 mL HBSS (with Ca/Mg) including DNAse I (10 mg/mL) and collagenase I (100 U/μL) at 37 °C and 350 rpm for 30 min (kidney) or 45 min (liver and lung). 2 ml PBS was added to the digestion buffer followed by filtering through a 70 μm cell strainer. Samples were then centrifuged at 400 g for 5 min

and cell pellets were resuspended in PBS with 1% FCS. Cells were incubated with antibody mixtures (1:100) for 30 min on ice, followed by fixation with 4% PFA for 10 min. RBC lysis was also required for the spleen tissue. Lineage markers were used for negative gating (Lin⁻). This included the markers (Ly6G, CD11b) and (CD19, NK1.1, TCRβ) for lymphoid and myeloid panels, respectively. Cells were washed with 2 ml PBS (400 g, 5 min) and resuspended in PBS with 1% FCS. All samples were acquired using BD FACSCanto II and the FACSDiva software 8.0 (BD Biosciences). Data analysis was performed using FlowJo v10.9. software (BD Bioscience). Fluorescence-activated cell sorting was used to extract viable CD45⁺ leukocytes from the blood, kidney microcirculation (CD45⁺CD45.2⁺), and kidney tissue (CD45⁺CD45.2⁻) as described above using a BD Aria II flow cytometer equipped with 11 detectors and FACSDiva software v9.0. Cells were sorted into tubes containing 200 μl PBS + 2% FCS on ice and washed with PBS once before further processing.

**Renal leukocyte enrichment**
For the cytokine assays, leukocytes were enriched in single cell suspensions using the MagniSort™ Mouse CD45 Positive Selection Kit (Thermo Fisher, cat. no. 8802-6865-74). The anti-CD45 antibody clone of this kit does not interfere with the anti-CD45 and anti-CD45.2

antibodies used for microvascular stainings. Cells were incubated with 20 μl of the anti-mouse biotinylated anti-CD45 antibody for 10 min at room temperature and washed with 2 mL PBS at 400 × g for 5 min. The cell pellet was resuspended in 500 μl of PBS with 1% FCS and incubated with 20 μl of streptavidin-coated magnetic beads for 10 min. 2 ml PBS were added and the tube was placed into the EasySep™ magnet (Stemcell Technologies, cat. no.18000) for 7 min. The supernatant was discarded, the sample tube was removed from the magnet and 2 ml of PBS were added. The magnet incubation steps were repeated two times. Cells were then washed with PBS (400 × g for 5 min) and resuspended in 1 ml cell culture medium (RPMI with 10% FCS).

### Intracellular cytokine analysis
100 μl of whole blood and the renal leukocyte cell suspension (after enrichment) were mixed with cell culture medium (RPMI1640 + 10% FCS) and incubated with LPS (kidney: 1.0 μg/ml, blood: 0.1 μg/ml) and 500x Protein Transport Inhibitor Cocktail (2 μl/ml) for 4 h at 37 °C and 5% $CO_2$. Unstimulated samples served as controls. After washing with 1x PBS, cell surface markers were stained for 30 min at 4 °C, followed by RBC lysis for blood samples. Intracellular staining of cytokines was performed by using the Foxp3 Transcription Factor Staining Set (eBioscience Foxp3/Transcription Factor Staining Buffer Set, no. 00-5523-00) according to the manufacturer's protocol. Briefly, cells were fixed with 300 μl of Foxp3 Fixation/Permeabilization working solution and incubated for 60 min at room temperature. Samples were washed twice with 1 ml of 1 × Permeabilization Buffer followed by a 30 min incubation at room temperature with the PE rat anti-mouse TNF-α antibody (Biolegend, cat. no. 506305, clone: MP6-XT22, dilution: 1:100). After two washing steps with 1x Permeabilization Buffer, cells were resuspended in PBS with 2% FCS and analyzed by flow cytometry. An isotype antibody was used as a control (PE Rat IgG1, κ Isotype Ctrl Antibody (Biolegend, cat. no. 400407, clone: RTK2071, dilution: 1:100).

### Phagocytosis assay
To determine the phagocytic activity of renal leukocytes, mice were i.v. injected with pHrodo deep red E. coli Bioparticles (1×10⁸ particles, Thermo Fisher, cat. no. P35360) 20 min prior to organ collection. These bioparticles become fluorescent in low pH environments such as endosomes or lysosomes, indicating cellular phagocytosis. Intravascular leukocyte staining was performed as described above. Kidneys were harvested for subsequent analysis by flow cytometry and immunofluorescence imaging. Mice without pHrodo E. coli injection were used as controls.

### Immunofluorescence microscopy
Freshly harvested tissue was cryopreserved in 10% sucrose solution overnight at 4 °C, embedded in Tissue Tek O.C.T (Sakura Finetek), frozen in liquid nitrogen and then stored at −80 °C. 7 μm sections were cut with a cryostat (Thermo Scientific, blade temperature: −25 °C, specimen temperature: −15 °C) and mounted on superfrost microscope slides. Sections were fixed with 4% paraformaldehyde (PFA) for 10 min, washed three times 5 min with PBS. For ex vivo staining, sections were blocked with 10% goat serum/5% bovine serum albumin (BSA) in PBS for 2 h at room temperature and then incubated with primary antibodies diluted in 5% goat serum/5% BSA in PBS overnight at 4 °C in a humidity chamber. After three washing steps, secondary antibodies diluted in 5% goat serum/5% BSA in PBS were applied on the sections for 2 h at room temperature. Cryosections were washed three times 5 min with PBS and were mounted with Fluoroshield with DAPI (Sigma Aldrich, F6057). Images were acquired using an Axio Imager M2 with Apotome 2.0 and 20x objective (Zeiss, Germany). For imaging of thick tissue sections, organs were fixed in 4% PFA overnight and were embedded using 3% low-melting agarose. 50−100 μm sections were cut with a vibratome (Leica, VT1000 S). For SDS delipidation, samples were incubated in a clearing solution (200 mM boric acid and 4% SDS,

pH 8.5) at 70 °C at 500 rpm for 1 h using a benchtop shaker. Free-floating sections were washed with PBS + 0,1 % Triton-X 100 (PBST) for 5 min at RT followed by incubation with primary conjugated antibodies in PBST overnight at 4 °C. After a further washing step with PBST, sections were mounted with Fluoroshield with DAPI onto microscope slides for confocal imaging (Leica, SP8 confocal microscope, 40x water objective). ImageJ/Fiji v2.15 (http://fiji.sc/) and Zen Software v3.8 (Zeiss, Germany) were used to analyze images. Cells counts were determined in microscopy images acquired using a 20x objective.

### Histology and scoring of kidney injury
Paraffin-embedded kidney sections were stained with hematoxylin and eosin (H&E) and periodic acid-Schiff (PAS) using standard protocols. Tubular injury based on casts, dilatation and damage was scored by two independent and blinded pathologists trained in nephropathology.

### Laboratory values
Heparinized cardiac blood was collected and centrifuged at 2000x g for 20 min. Plasma was transferred into cryotubes and stored at −20 °C. Creatinine and blood urea nitrogen as markers for AKI were measured by IDEXX Laboratories (Kornwestheim, Germany).

### Targeted RNA single-cell library preparation and sequencing
For single-cell RNA sequencing, the BD Rhapsody Express System (BD Biosciences) was used according to the manufacturer's protocols. Viable CD45⁺/CD45.2⁺ (microcirculation), CD45⁺/CD45.2⁻ (tissue) and CD45.2+ peripheral blood leukocytes were FACS-sorted and labeled using the BD Rhapsody Single-Cell Multiplexing CTT Kit (BD Biosciences, 626545). The sorted populations were > 95% pure as determined by sorting experiments with a post-sort purity control by flow cytometry. The cell numbers for each mouse and multiplexed subset ranged between 571 and 5533 cells (median: 2422 cells (untreated), 3798 cells (AKI-reg)).

Sample tags were incubated at room temperature (RT) for 20 min and washed three times. The pooled sample was resuspended in an ice-cold sample buffer (BD Biosciences, BD Rhapsody Cartridge Reagent Kit, 633731). Isolation of single cells was performed using the single-cell capture and cDNA synthesis kit according to the manufacturer's protocol. Briefly, the samples were loaded onto the primed nanowell cartridge (BD Biosciences, 633733), incubated at room temperature for 15 min followed by cell capture beads and further incubation (3 min at room temperature). Cells were lysed followed by bead retrieval and washing. Reverse transcription was performed for cDNA synthesis following Exonuclease I treatment (BD Biosciences, 633773).

Libraries were prepared using the BD Rhapsody targeted mRNA and Abseq amplification kit (BD Biosciences, 633774). Targeted amplification of cDNA was performed using a mouse immune response panel (397 genes, BD Biosciences, 633753) and a custom additional panel of 99 genes (covering leukocyte lineage- and kidney cell-specific genes, supplementary Data 3) by PCR (11−15 cycles). For double-sided DNA size selection, Agencourt AMPure XP magnetic beads (Beckman Colter, A63880) were used to separate sample tag PCR products from mRNA target PCR products. Further amplification of the sample tag and the mRNA-targeted PCR products was performed by PCR (10 cycles). The PCR products were then purified using Agencourt AMPure XP magnetic beads. The concentration of each sample was determined using a Qubit fluorometer and the Qubit dsDNA HS Assay Kit (Thermo Fisher Scientific, Q32851). To prepare the final libraries, the purified PCR products were indexed by PCR (6−8 cycles). The index PCR products were purified using Agencourt AMPure XP magnetic beads. Quality control was performed by measuring the concentration and the average fragment size of the mRNA target library and sample tag library using the Agilent Tape Station with the High Sensitivity D1000 ScreenTape (Agilent, 5067-5584). The final libraries were diluted to a

concentration of 4 nM and multiplexed for Novaseq paired-end sequencing (150 bp) including 20% PhiX spike-in. The sequencing depth was 600 and 4000 reads/cell for the sample tag and the mRNA library, respectively. Sequencing was performed using the Illumina NovaSeq 6000 sequencer.

## Sequencing data analysis

Demultiplexing and preprocessing of the sequencing data were conducted with the BD Rhapsody sequence analysis pipeline v.1.11 from SevenBridges according to the manufacturer's protocol. Sequences were aligned against a targeted genome panel based on the murine reference mm10. All error-corrected scRNA count matrices were imported into R v4.0.5 and Seurat v4[41]. For each sample, only cells with at least five features were considered, and rare features that were present in less than 10 cells per sample were discarded. To remove potential doublets or multiplets from the analysis, cells with a very high RNA count value were filtered, with thresholds ranging from 2.000 to 10.000 depending on the sample's nCount_RNA value distribution. Subsequently, Seurat's SCTransform routine was used to normalize and integrate the scRNA samples for each data set, while PCA and UMAP were employed for dimension reduction. Finally, clusterings for each data set were created with Seurat's FindClusters function, using a resolution of 0.5 and default values otherwise. Additionally, all identified B and macrophages/DC were extracted and reclustered with a resolution parameter of 0.2 and 0.15, respectively, to allow for a more detailed view.

For all data sets and subsets, clusters were annotated using a combined approach of marker gene visualization and differential gene expression (non-parametric Wilcoxon rank sum test). The top 10 differentially expressed genes for all clusters in all data sets are listed in supplementary Data 2. Target genes were selected and expression patterns were visualized with Seurat's FeaturePlot functionality. The package's FindMarkers routine was chosen to identify differentially expressed genes per cluster using the MAST algorithm[42]. Pseudotime and trajectory analysis was conducted with the R/Bioconductor package Slingshot v1.8[43]. Briefly, Seurat cell embeddings were imported and lineages were calculated based on the chosen start cluster of interest (the main B cell cluster present in peripheral blood). Pseudotime and tree visualizations were then realized with Slingshot's plot functions.

## Statistics and study design

Statistical tests are indicated in the figure legends. All $t$ tests are two sided. Mean and standard deviation (SD) are visualized. Significance levels: $****p < 0.0001$, $***p < 0.001$, $**p < 0.01$, $*p < 0.05$. Investigators were not blinded to the experimental conditions due to technical reasons. Control experiments were run simultaneously to avoid a technical bias. Sample sizes for each method were mostly determined based on experience and published evidence. Each experiment included at least 3 independent biological replicates.

## Reporting summary

Further information on research design is available in the Nature Portfolio Reporting Summary linked to this article.

## Data availability

The sequencing data generated in this study have been deposited in the GEO database under accession code GSE252496. All other data are available in the article and its Supplementary files or from the corresponding author upon request. Source data are provided with this paper.

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

## Acknowledgments

The authors thank Oliver Söhnlein for helpful discussions. We thank A. Rosemann for technical support. We thank the team of the Molecular Nephrology lab and the Institute of Musculoskeletal Medicine for their support. This work was funded by the IMF University of Münster to KB (BU111801 and BU122006) and Deutsche Forschungsgemeinschaft (DFG) to KB (BU3247/6-1). The icons in Fig. 1a were used from Servier Medical Art (https://smart.servier.com) under CC BY 4.0.

## Author contributions

R.R. performed most experiments and analyzed data. P.S. established and performed sequencing experiments and analyzed data. C.W. and J.V. performed most bioinformatic analyses. BHe and V.M. graded histopathological injury patterns. G.V. performed flow cytometry experiments. BHu performed immunofluorescence microscopy. D.F., T.P. and H.P. advised on main concepts, experiment design and data analysis. K.B. designed the project, analyzed data, supervised experiments, and wrote the manuscript.

## Funding

## Competing interests

The authors declare no competing interests.

## Additional information

**Publisher's note** Springer Natur+e remains neutral with regard to jurisdictional claims in published maps and institutional affiliations.

