## [Transparent Peer Review file · Nature Communications]

Microvascular immunity is organ-specific and remodeled after kidney injury

Corresponding Author: Dr Konrad Buscher

Version 0:

Reviewer comments:

Reviewer #1

(Remarks to the Author)

The authors provided several scRNAseq datasets and described differences among circulating, microvascular and tissue immune cells in the healthy, diseased and regenerated context. Their aim was to identify inherent immunological parameters of leukocytes in relation to this spatial information. They conclude "Collectively, our findings suggest a paradigm of organ- and disease-specific microvascular immunity that largely eludes conventional blood and tissue analysis." (Abstract) Their findings are novel and interesting, but remain descriptive: The functional differences between the subsets in the blood, microvasculature or tissue was not addressed, nor were the biological reasons resulting in such differences addressed. Furthermore, there are concerns regarding the validity of their findings and about the conclusiveness of their main claim.

Main concerns:

1. Their microvascular cells seem to comprise both intravascular cells and vessel attached cells, given they did not perfuse the organs. They are aware of this and state in the methods "To account for this effect, we used peripheral blood as a reference." Then they exclude in Figure 2 from the tissue cells the cells with the genomic fingerprint of blood cells and consider the remaining cells "microvascular cells". Their main conclusion in the abstract, i.e. that microvascular cells differ from blood cells, thus seems to be a self-fulfilling prophecy: If one excludes from a group A of cells those that resemble cells of a group B, then of course the cells remaining in A (here "microvascular cells") differ from the cells in group B (blood). This is circular reasoning!
2. They showed there was no leaking CD45.2 antibody within the kidney and lung in 5min, what about liver and spleen? The leaking time likely differs between organs.
3. Several cell subsets were changed in the blood or microvasculature under disease and regenerative context compared to healthy mice. What is the function of those cell subsets and their contribution to the progression of disease and the regenerative process? Which subset is essential for the development of disease or regenerative process in tissue?
4. The authors observed abundant F4/80+ CD11c+ macrophages in the kidney microvasculature, almost 100 fold higher than in the tissue. It is hard to believe that kidney resident macrophages, which are F4/80+ and CD11c+, are mostly microvascular. They did not confirm this claim by flow cytometry nor histology
5. The authors found a unique endovascular B cell subset in kidney. Did they confirm the existence of this subset by flow cytometry or histology? Does it exist in other organs? What is the function of this subset?
6. The Slingshot figure 3 lacks a direction sign of the development course, making it difficult to interpret the results. Upregulation of Egr1 and Irf4 based on the figure 3d were unclear.

Minor concerns:

7. In the barplot of fig1b, as the author would like to compare the microvasuclar and tissue cells in each organ, it is better to put them in the same barplot and mark the statistical significance.
8. The color scale of fig3d is missing
9. In fig4b, why was there a high fraction of undefined cell in microvasculature? what is this cell subset?

10. The authors sequenced 33,474 leukocytes, including 13,828, 13,408, and 6,238 in blood, microcirculation, and kidney, respectively. This distribution ratio is close to 2:2:1, indicating a potential bias against the kidney in the single-cell experiment. The small number of cells might bias the bioinformatic conclusions drawn.

11. the authors chose two disease models, one of which is semi-sterile peritonitis. This is not directly related to any of the organs studied. Its direct impact on organ immunity cannot be assessed. The authors should explain their rationale better.

12. Figure 3e: cluster 11 seems as good as cluster 0. The authors should explain why not use cluster 11? so why not cluster 2+11 vs. cluster 4+7?

13. The scale information needs to be clearly shown in Figure 6d.

14. The authors concluded that after renal recovery, most cellular alterations were in the microvascular niche, and none could be detected in the peripheral blood. However, as the authors themselves stated, they only provided a snapshot in time during the whole recovery phase. Therefore, caution should be exercised when drawing general conclusions.

15. Can they speculate on the mechanistic connection between microvascular cells and tissue cells? How does tissue inflammation influence the microvascular cells?

Reviewer #2

(Remarks to the Author)

These experiments address the critical clinical question of whether analytics performed on cells in the peripheral circulation accurately reflect resting or inflammatory conditions in tissues. It is known that non-circulating or “resident” immune cells are integral components of barrier tissues. With the application of single cell RNA sequencing, tissue atlases have been established that define transcript signatures for resident immune cells in many human and mouse organs including the kidney and lung (Lake BB, et al.2023; Zimmerman KA, et al.2019). The current study expands this concept by adding another stratum, namely the contents of the microvasculature in kidney, liver and lung both in the steady state and during two forms of inflammation. The concept of leukocytes differentially marginating in the microvasculature is not new, but the use of single cell RNA sequencing to define these cells could add new insights.

In this study, a surprisingly large number of B cells were identified in the microvasculature of the healthy kidney. This is a novel and potentially mechanistically important finding. Primarily located in glomerular capillaries, these “endovascular B cells” accounted for 86% of the renal B cells. These cells are variously characterized as Egr1+ CD69+ CXCR4+ B cells in one supplementary table; Fcγ2a+Ccr7+ in Figure 2e; and subsequently subcategorized into 3 major subsets: the most microvascular exclusive one defined by Myc and CD69, the second by Erg1 and Dusp 1+2, and the third by Irf4, Cxcr4 and Ccr7 (Figure 3a). Slingshot trajectory analysis was performed to indicate probable phenotypic transformations from peripheral blood B cells to these microvascular subcategories. It is not discussed whether these are terminal differentiations or whether there are possible switches between these subpopulations. For example, it appears in Figure 3d that cluster 7, the most microvascular exclusive subpopulation, is on the same trajectory as cluster 4, the second most microvascular exclusive subpopulation. Does this indicate that B cells in cluster 4 leak back into the circulation?

The dynamics of the microvascular populations were tested in models of acute kidney injury (produced by temporary clamping of the renal pedicles) and “semi-sterile” peritonitis (laparotomy with gut mobilization) both in the acute phase (day 1) and in recovery (day 12). Acute kidney injury (AKI) caused the endovascular B cells, which according to Figure 2c and Supplemental Fig 6 (Cluster #3) are highly numerous in the kidney, to decrease acutely in the microvasculature of the kidney as well as the lung, liver and blood (Suppl Fig 4). The investigators do not account for where these cells relocate, although it could be surmised from the graph of tissues in main Fig 4 that the B cells might migrate to the spleen. During recovery (day 12) from AKI, one cluster of B cells (#3) remains downregulated (Fig 5f and Supplemental Fig 6). This cluster is annotated in the supplementary table simply as naïve B cells with no distinction from the other 2 clusters of B cells (#1 & 5). The genetic signatures that distinguish these clusters would be worthwhile documenting.

As expected neutrophils increased in the renal microvasculature during the acute phase of AKI and decreased in recovery, whereas the nonclassical (Ly6C^{lo}) monocytes increased during recovery. These changes were largely reflected in the blood and also pulmonary microvasculature.

Additional points:

1. The most notable discrepancy between cells in the peripheral blood and microvasculature in Figure 4d was the side scatter (SSC) high monocytes that the investigators describe only as being “more granular and represent a macrophage-like phenotype”. No additional stains or data are present to support this designation.

2. The designation of monocytes in the flow cytometry data is concerning. These cells are depicted as being Lin⁻ (what antigens make up Lin⁻ for this condition is not made clear in the main figure, legend, supplement, legend or the methods section), CD11b⁺ and Ly6C^{lo}. The data in Figure 1D very nicely shows, with in vivo and ex vivo CD45 staining, that some cells in the tissue project out into the lumen of the microvasculature. This could be due to tissue resident dendritic cells many of which express CD11b (Mesnil, et al. 2012). The presence of these cells is supported by the scRNAseq data that clearly define cDC2 populations. Without including CD11c in the flow cytometry panel, DCs cannot be excluded from monocytes.

3. The methodology and use of i.v. CD45 antibodies for labelled is aligned with the field, and is common practice in studies focused on peripheral vs tissue resident immune cells. Though there is a pitfall in this methodology. As the authors note (Yatim, et al. 2016; Vollmann, et al. 2021), there are cells that extend protrusions out into the microvascular lumen, these cells would be labeled in vivo by i.v. CD45 antibodies, as such these cells will provide a false positive in the cells designated as cells of the microvasculature (Figure 1d) skewing the data and any possible interpretations. This caveat should be discussed.

4. Although data are presented for splenic microvasculature, the concept of microvasculature and the technique of pulse labeling being confined to the vasculature is questionable for spleen with about 90 percent of the blood flowing through an open sinusoidal route of circulation.

5. There is a disconnect between the text references to supplemental tables. The text refers to supplemental table 1 as listing the 99 custom additional added genes and then on the same page as listing 27 CD45⁺ leukocyte clusters. No other

tables are referenced. Yet there is a supplemental table with 26 leukocyte clusters and another with 29 leukocyte clusters.

Overall, this is a highly observational set of experiments. The data are presented unsystematically and it is difficult to piece together continuous paths of given cell types. For example, the large population of potentially novel endovascular B cells in healthy kidneys is not discussed relative to the models of inflammation. Insights about any possible functional significance of the novel finding of endovascular B cells and their responses to AKI would be valuable especially in the light of recent evidence that CCL7 producing B cells mediate ischemia-reperfusion injury in the lung (Farahnak, et al. 2024). More comprehensive characterization of the endovascular B cells might allow the investigators to use depleting antibodies or other approaches to intervene in models of AKI and provide evidence of causality.

References:

Lake BB, et al. An atlas of healthy and injured cell states and niches in the human kidney. *Nature* 619, 585-594 (2023).

Zimmerman KA, et al. Single-Cell RNA Sequencing Identifies Candidate Renal Resident Macrophage Gene Expression Signatures across Species. *J Am Soc Nephrol* 30, 767-781 (2019).

Mesnil C, et al. Resident CD11b(+)Ly6C(-) lung dendritic cells are responsible for allergic airway sensitization to house dust mite in mice. *PLoS One* 7, e53242 (2012).

Farahnak K, et al. B cells mediate lung ischemia/reperfusion injury by recruiting classical monocytes via synergistic B cell receptor/TLR4 signaling. *The Journal of Clinical Investigation* 134, (2024).

Reviewer #3

(Remarks to the Author)

General Comments: This manuscript addresses a very interesting question regarding what we are measuring when we analyze leukocyte populations isolated from various organs. The premise of the paper is that cells in the microcirculation are not equivalent to those in the peripheral blood, with the latter defined as blood obtained from a major vessel or from a cardiac ventricle. The methodology employed is entirely dependent on the veracity of the method used to label intravascular cells. The authors validated the technique by injected antibody specific for an epithelial marker and finding no staining in the extravascular space. The extravascular leukocytes, then, are identified by staining with CD45.2 anti-allotypic antibody and comparing with CD45 total staining. The cell types were then separated using flow cytometric sorting. If this methodology is as discriminatory as it appears to be, then the data indicate a variable in evaluation of intraorgan leukocytes that has not been previously considered in this manner. While this reviewer might disagree with some of the minor details, the general conclusions are well supported by the data and warrant further study.

Specific comments:

1. Using a gradient for isolation of cells can significantly skew the results. The authors avoided this common pitfall.
2. Paragraph starting with: "To detect unsupervised phenotypes and transcriptional states of microvascular leukocytes..." How the data were multiplexed and processed could be clarified. It seems the data are derived from a total of four mice and the data from each mouse was multiplexed with renal tissue leukocytes, microcirculation and peripheral blood? For tissue, then, about 1500 leukocytes were sequenced per kidney. Were the contributions from each mouse roughly equal?
3. It is stated that Adgre1+Itgax+Fcgr4+ macrophages in the kidney were almost entirely microvascular. The data in figure 2e (cluster 0) do not support that interpretation because the difference (in proportion) are not statistically significant. In considering proportion, the figure does not state what the denominator is – CD45+ total? Also, if one is to make such a statement, then the data to be compared should be absolute numbers.
4. For the pseudotime measurement, it would be helpful to the reader to specify how the initial point was chosen.
5. Paragraph: "Myeloid cells are major effectors in ischemic AKI". It is stated "phagocytes were absent in the peripheral blood...". Phagocytes is a vague term in this context and essentially encompasses almost all myeloid cells. Is that the intended meaning?

Reviewer #4

(Remarks to the Author)

Version 1:

Reviewer comments:

Reviewer #1

(Remarks to the Author)

The authors have addressed almost all of my concerns satisfactorily. Only the first point needs some more editing: They responded to it:

“We believe that this is a misunderstanding. Two cell populations (or groups) were separated from the kidney: microvascular (CD45+CD45.2iv+) and tissue (CD45+CD45.2iv-) leukocytes. The former group was then compared to a third group, namely peripheral blood, that was separately collected from the right heart (figure below).”

The authors did not get my point: I pointed out that their microvascular cells must contain some vascular cells, ie cells in large vessels located within the kidney. These are not in the microcirculation, but in essence equivalent to the cells in their third group, the blood. Fig 2c shows quite clearly overlap between several clusters of blood and their microvascular cells, especially in monocytes, B and T cells, ie the cells expected to be present within vessels. Thus, not all their microvascular cells are microvascular cells; some are in fact blood cells. Those that differ in scRNASeq from blood cells will of course show differences from blood cells, by definition. They have removed this circular reasoning from their paper and focussed their manuscript on cells present only in the microcirculation. I am satisfied with this. But I would recommend to more clearly mention that this contamination exists and discuss it.

Reviewer #2

(Remarks to the Author)

The investigators have extensively revised the text, performed additional experiments and added multiple new figures including a graphical summary in response to the reviews.

Specifically, they have added a bar graph depicting an ontological analysis of microvascular B cells (figure 3f). The investigators forego further analysis of B cells because “tools to target intravascular B cells specifically are still lacking, and therefore, the precise functions and mechanisms remain unknown.” Instead, they have chosen to focus on macrophages and DCs. To this end, they have further delineated the cell subsets of the microvascular SSChi cluster using F4/80, CD11c and CD11b, and represented the data in a stacked bar graph (figure 1e). They also reanalyzed the macrophage and DC RNAseq data (figure 6) as well as performed a functional assay of E Coli phagocytosis (Figure 7).

While these changes provide some added information, the paper primarily presents a catalogue of data and misses opportunities to make a cohesive story.

For example, in figure 6 the subclustering of macrophages and DCs is presented merely by multiple different graphics illustrating increase or decrease relative to perturbation and location. The authors present data in figure 6c, e and f indicating that cluster 2 is increased 12 days after AKI in the microvasculature. Instead of using their own data to determine what genes might be differentially expressed in the cluster in response to AKI, they perform flow cytometry using “the common markers CD11c and F4/80” in figure 6h. The CD11c+F4/80+ subpopulation in flow is somewhat counterintuitive given that cluster 2 has low expression of Adgre1 (F4/80) and high expression of Itgax (CD11c), whereas clusters 0,1 and 6 have high expression of Adgre1 (F4/80) and are low expressors of Itgax (CD11c). Similarly, the identification of the phagocytic cells in figure 7 could be better integrated with the clusters defined in figure 6 with a more judicious choice of cell markers.

In presenting the data in figure 6, the authors attribute function based on a selected few genes, such as cluster 0 “showed a fibrosis-related gene signature with upregulation of Col14a1 and Tgfb3 (figure 6b).” This conclusion would be better supported with a gene ontology analysis. The authors also interpret findings for perivascular C1q+ macrophages as indicating a retraction from the endoluminal space (figure 6e,f). The word retraction or retract is used several times in this context and it is not clear whether the authors mean that these cells retract intraluminal extensions, which could be documented by histological means as in figure 1d.

The graphical summary in figure 7 is helpful. However, the cluster nomenclature can lead to some confusion. For example, cluster 0 in the left panel is not the same as in the right panel. Possibly the cluster designations could be omitted.

Finally, the new title that indicates the findings demonstrate the changes microvascular cell content “maladapts after kidney injury” is not supported by any functional data.

Reviewer #3

(Remarks to the Author)

I am satisfied with the changes made to the manuscript.

Reviewer #4

(Remarks to the Author)

Version 2:

Reviewer comments:

Reviewer #2

(Remarks to the Author)

The authors have made extensive responses to our critiques. The accounting of the authors' attempts to find unique surface marker combinations to exclusively label cells in cluster 2 is informative. We also appreciate the multiple other thoughtful clarifications and revisions of the text including a more precise title.

Reviewer #4

(Remarks to the Author)

#Point to point response

Revision of the manuscript NCOMMS-24-03267, "Microvascular immunity is organ-specific and maladapted after kidney injury", Rixen R et al., Nov 17, 2024

Reviewer #1

The authors provided several scRNAseq datasets and described differences among circulating, microvascular and tissue immune cells in the healthy, diseased and regenerated context. Their aim was to identify inherent immunological parameters of leukocytes in relation to this spatial information. They conclude "Collectively, our findings suggest a paradigm of organ- and disease-specific microvascular immunity that largely eludes conventional blood and tissue analysis." (Abstract)

Their findings are novel and interesting, but remain descriptive: The functional differences between the subsets in the blood, microvasculature or tissue was not addressed, nor were the biological reasons resulting in such differences addressed. Furthermore, there are concerns regarding the validity of their findings and about the conclusiveness of their main claim.

We thank the reviewer for the time, thorough review of our manuscript and many suggestions to improve our work. We have now added functional data about microvascular phagocyte subsets including M2 polarization, phagocytosis and TNF secretion (new main figures 6 and 7). Our data highlight a new role of microvascular immune cells in maladaptive repair after AKI. These findings could provide new insights into the clinical observation that AKI is a risk factor for recurrent AKI that affects up to 30% of AKI patients.

We also added another experiment to validate our methodological approach (suppl. fig 1).

Main concerns:

1. Their microvascular cells seem to comprise both intravascular cells and vessel attached cells, given they did not perfuse the organs. They are aware of this and state in the methods "To account for this effect, we used peripheral blood as a reference." Then they exclude in Figure 2 from the tissue cells the cells with the genomic fingerprint of blood cells and consider the remaining cells "microvascular cells". Their main conclusion in the abstract, i.e. that microvascular cells differ from blood cells, thus seems to be a self-fulfilling prophecy: If one excludes from a group A of cells those that resemble cells of a group B, then of course the cells remaining in A (here "microvascular cells") differ from the cells in group B (blood). This is circular reasoning!

We believe that this is a misunderstanding. Two cell populations (or groups) were separated from the kidney: microvascular (CD45+CD45.2iv+) and tissue (CD45+CD45.2iv-) leukocytes. The former group was then compared to a third group, namely peripheral blood, that was separately collected from the right heart (figure below). We changed the first paragraphs of the result section to better explain the experimental approach.

2. They showed there was no leaking CD45.2 antibody within the kidney and lung in 5min, what about liver and spleen? The leaking time likely differs between organs.

We now show the corresponding data in the liver. Here, EpCAM-1 (which was used in kidney and lung) is not expressed in the tissue. Therefore, we used E-cadherin as an epithelial marker. After iv-injection and a circulation time of 5 min, the extravascular stainings were detected using flow cytometry. Ex vivo stainings of E-cadherin and isotype antibodies were used as controls. These experiments were performed in untreated, AKI and recovery after AKI (AKI-reg) conditions. Kidney, lung, liver and spleen were investigated. The data confirm that the intravascular antibody does not stain extravascular epitopes using our protocol (data shown in suppl. fig 1). However, the spleen is a notable exception. The terminal bed of the splenic vasculature is made of sinusoids with fenestrated endothelium. Intravascular CD45 antibodies stain sinusoidal leukocytes in the marginal zone as shown by immunofluorescence images (figure 1, supplemental figure 3). Since the spleen is negative for E-cadherin and EpCAM-1, and lacks a comparable vascular barrier, we cannot draw similar conclusions. This caveat is now more emphasized in the manuscript (first paragraph of the result section and paragraph "limitations" in the the discussion).

3. Several cell subsets were changed in the blood or microvasculature under disease and regenerative context compared to healthy mice. What is the function of those cell subsets and their contribution to the progression of disease and the regenerative process? Which subset is essential for the development of disease or regenerative process in tissue?

We now show new data about microvascular phagocyte functions of the kidney (new main figures 6 and 7). In these experiments, using the intravascular labeling approach, we can directly compare tissue resident and microvascular macrophages and dendritic cells. We observed a breakdown of the microvascular interface after AKI. The large population of homeostatic C1q+ macrophages retract and accumulate in the tissue, while up to 4 new macrophage/DC populations infiltrate the microvascular interface after AKI (figure 6). Microvascular (but not tissue-resident) F480+CD11c+ macrophages become M2 polarized and increase their capacity to phagocytose blood borne bacteria (iv injected pHrodo E.coli bioparticles). In addition, TNF secretion (baseline and in response to bacterial LPS) is increased after AKI particularly in microvascular CD11c+F480- dendritic cells while being absent in tissue resident counterparts.

DCs have been identified as main TNF producing cells (Dong X et al, Kidney Int 2007). TNF is known to trigger neutrophil influx and renal tubular epithelial cell apoptosis in ischemic AKI further exacerbating kidney injury (Donnahoo KK et al, Am J Physiol 1999; Misseri R et al. Am J Physiol Renal Physiol., 2005). Our data suggest that TNF-producing DC predominantly occupy the microvascular niche after AKI regeneration. Therefore, we interpret our findings

as microvascular maladaptions after injury. These findings could provide new insights into the clinical observation that AKI is a risk factor for recurrent AKI that affects up to 30% of AKI patients.

A graphical summary is now added in figure 7:

4. The authors observed abundant F4/80+ CD11c+ macrophages in the kidney microvasculature, almost 100 fold higher than in the tissue. It is hard to believe that kidney resident macrophages, which are F4/80+ and CD11c+, are mostly microvascular. They did not confirm this claim by flow cytometry nor histology

Our finding of a 100-fold upregulation is based on the ratio of microvascular to blood leukocytes. Because these cells are rarely detectable in the peripheral blood, the ratio is very high. When comparing microvascular to tissue leukocytes, the ratio is different: The scRNAseq data suggests that 67% of all renal macrophages and dendritic cells localize in the microvasculature (figure 6). Staining CD11c by intravenous injection and ex vivo in immunofluorescence microscopy, we find that about 35-40% of all renal CD11c+ cells are microvascular (figure 6).

5. The authors found a unique endovascular B cell subset in kidney. Did they confirm the existence of this subset by flow cytometry or histology? Does it exist in other organs? What is the function of this subset?

We identified this subset by scRNAseq in the microvasculature of the kidney, and found them to be enriched in glomerula using fluorescence microscopy (figure 3). It can be hypothesized that glomerular capillaries allow circulating B cells to adhere and interact. This validates earlier reports on intravascular B cells of the heart (Adamo L et al, JCI Insight, 2020; Bermea KC et al, Frontiers in Immunology, 2022). Adamo et al showed that systemic B cell depletion altered the myocardial leukocyte composition, suggesting a role in leukocyte trafficking. In addition, adoptive transfer of peripheral B cells in B cell-deficient mice replenished intravascular B cells of the heart (Adamo L et al, JCI Insight). However, tools to target organ-specific and intravascular B cells specifically are still lacking, and precise functions remain unknown.

The reviewer's questions about B cells are very relevant. In fact, many of the newly discovered microvascular cell types require further investigation including B cells, ILC2 and

plasmacytoid DCs. However, we decided to focus on renal macrophages and dendritic cells because they are well known to play a pivotal role in ischemic kidney injury (Maryam B, Kidney360, 2024; Kurts C, Nat Rev Nephrology, 2020). Our new data in figure 6 and 7 analyze the structural breakdown of the microvascular interface after injury in detail and highlight the infiltration of new macrophages and DCs with specific functions.

6. The Slingshot figure 3 lacks a direction sign of the development course, making it difficult to interpret the results. Upregulation of Egr1 and Irf4 based on the figure 3d were unclear.

We added the direction of the trajectory in figure 3. Cluster 0 was selected as origin because it represents the main (and largest) systemic B cell cluster which is significantly enriched in the peripheral blood (see figure 3c). Slingshot calculated transcriptomic trajectories to other cell clusters and found two trajectories that directly interconnect the microvascular and blood B cell populations. The main gene expression changes along these trajectories were Irf4 and Egr1.

Minor concerns:

7. In the barplot of fig1b, as the author would like to compare the microvasuclar and tissue cells in each organ, it is better to put them in the same barplot and mark the statistical significance.

We changed the barplot as suggested.

8. The color scale of fig3d is missing

The color scale was added.

9. In fig4b, why was there a high fraction of undefined cell in microvasculature? what is this cell subset?

We thank the reviewer for pointing this out and apologize for this mistake. We reanalyzed the data and found a technical error. The myeloid and lymphoid marker panels were used separately in flow cytometry experiments. The non-myeloid cell fraction in the myeloid panel (and vice versa) was labeled "undefined" when joining both data sets. We now corrected the graph in figure 4b.

10. The authors sequenced 33,474 leukocytes, including 13,828, 13,408, and 6,238 in blood, microcirculation, and kidney, respectively. This distribution ratio is close to 2:2:1, indicating a potential bias against the kidney in the single-cell experiment. The small number of cells might bias the bioinformatic conclusions drawn.

We re-analyzed our data with a downscaled data set (1:1:1) and confirmed the results shown in figure 2. This control is now mentioned in the method section.

11. the authors chose two disease models, one of which is semi-sterile peritonitis. This is not directly related to any of the organs studied. Its direct impact on organ immunity cannot be assessed. The authors should explain their rationale better.

We chose the peritonitis model because it also serves as a control experiment for ischemic AKI. By opening the abdomen and moving the intestines without clamping the kidney pedicles, the effects of the semi-sterile peritonitis can be distinguished from the effects of ischemic AKI. This allowed us to determine that the microvascular presence of neutrophils, classical and nonclassical monocytes in the liver is specific for AKI-reg and not related to the peritonitis (figure 4). We edited the manuscript to better explain this background.

12. Figure 3e: cluster 11 seems as good as cluster 0. The authors should explain why not use cluster 11? so why not cluster 2+11 vs. cluster 4+7?

We have changed the analysis in figure 3 in order to include only significant changes in the analysis. The significance for each cluster is now visualized in figure 3d. Clusters 0 and 5 are significantly downregulated (i.e. more present in blood), while cluster 1, 6, 4 and 7 are significantly upregulated (i.e. more present in the renal microvasculature). We then repeated the function enrichment analysis in figure 3f using the differentially expressed genes of these clusters. The results are partially different compared to the previous analysis due to the larger gene input list. We picked the GO terms that show the largest difference in enrichment between blood borne and microvascular B cell clusters (figure 3f).

13. The scale information needs to be clearly shown in Figure 6d.

The previous figure 6 was removed and replaced by figure 6 and 7 in order to focus on the phenotype and function of microvascular macrophages and DC.

14. The authors concluded that after renal recovery, most cellular alterations were in the microvascular niche, and none could be detected in the peripheral blood. However, as the authors themselves stated, they only provided a snapshot in time during the whole recovery phase. Therefore, caution should be exercised when drawing general conclusions.

We rephrased the text to avoid general conclusions.

15. Can they speculate on the mechanistic connection between microvascular cells and tissue cells? How does tissue inflammation influence the microvascular cells?

These are interesting questions. Marginated, perivascular and intravascular leukocytes occupy different positions of the blood-kidney barrier, likely affecting their ability to interact with their environment. The microanatomy also changes along the microvasculature from arterioles to capillaries and venules including the endothelial cell layer, the basement membrane or perivascular cell types. In addition, our data suggest microvascular leukocytes to be organ- and disease-specific. Therefore, we are cautious to speculate on specific functions of these cells at this moment. Further investigations are needed. Our manuscript aims to highlight the complexity and dynamics of the microvascular interface that are often not detectable in the peripheral blood or the tissue compartment. We hope our work will spark new interest in microvascular immunity across many different disciplines.

Reviewer #2 (Remarks to the Author):

These experiments address the critical clinical question of whether analytics performed on cells in the peripheral circulation accurately reflect resting or inflammatory conditions in tissues. It is known that non-circulating or “resident” immune cells are integral components of barrier tissues. With the application of single cell RNA sequencing, tissue atlases have been established that define transcript signatures for resident immune cells in many human and mouse organs including the kidney and lung (Lake BB, et al.2023; Zimmerman KA, et al.2019). The current study expands this concept by adding another stratum, namely the contents of the microvasculature in kidney, liver and lung both in the steady state and during two forms of inflammation. The concept of leukocytes differentially marginating in the microvasculature is not new, but the use of single cell RNA sequencing to define these cells could add new insights.

In this study, a surprisingly large number of B cells were identified in the microvasculature of the healthy kidney. This is a novel and potentially mechanistically important finding. Primarily located in glomerular capillaries, these “endovascular B cells” accounted for 86% of the renal B cells. These cells are variously characterized as Egr1+ CD69+ CXCR4+ B cells in one supplementary table; Fcεr2a+Ccr7+ in Figure 2e; and subsequently subcategorized into 3 major subsets: the most microvascular exclusive one defined by Myc and CD69, the second by Erg1 and Dusp 1+2, and the third by Irf4, Cxcr4 and Ccr7 (Figure 3a). Slingshot trajectory analysis was performed to indicate probable phenotypic transformations from peripheral blood B cells to these microvascular subcategories. It is not discussed whether these are terminal differentiations or whether there are possible switches between these subpopulations. For example, it appears in Figure 3d that cluster 7, the most microvascular exclusive subpopulation, is on the same trajectory as cluster 4, the second most microvascular exclusive subpopulation. Does this indicate that B cells in cluster 4 leak back into the circulation?

The dynamics of the microvascular populations were tested in models of acute kidney injury (produced by temporary clamping of the renal pedicles) and “semi-sterile” peritonitis (laparotomy with gut mobilization) both in the acute phase (day 1) and in recovery (day 12). Acute kidney injury (AKI) caused the endovascular B cells, which according to Figure 2c and Supplemental Fig 6 (Cluster #3) are highly numerous in the kidney, to decrease acutely in the microvasculature of the kidney as well as the lung, liver and blood (Suppl Fig 4). The investigators do not account for where these cells relocate, although it could be surmised from the graph of tissues in main Fig 4 that the B cells might migrate to the spleen. During recovery (day 12) from AKI, one cluster of B cells (#3) remains downregulated (Fig 5f and Supplemental Fig 6). This cluster is annotated in the supplementary table simply as naïve B cells with no distinction from the other 2 clusters of B cells (#1 & 5). The genetic signatures that distinguish these clusters would be worthwhile documenting.

As expected neutrophils increased in the renal microvasculature during the acute phase of AKI and decreased in recovery, whereas the nonclassical (Ly6Cl^o) monocytes increased during recovery. These changes were largely reflected in the blood and also pulmonary microvasculature.

We thank the reviewer for the time, thorough review of our manuscript and many suggestions to improve our work.

The reviewer's questions about B cells are very relevant. We had conducted an in-depth analysis of the B cell clusters because these cells are exclusively located in the microvasculature while being absent in the peripheral blood and the tissue. This contrasts with most other microvascular cell types in the kidney, that are also present in the tissue, suggesting a perivascular localization with intraluminal extensions.

This finding validates earlier reports on intravascular B cells of the heart (Adamo L et al, JCI Insight, 2020; Bermea KC et al, Frontiers in Immunology, 2022). Here, systemic B cell depletion altered the myocardial leukocyte composition, suggesting a role in microvascular leukocyte trafficking. However, tools to target intravascular B cells specifically are still lacking, and therefore, the precise functions and mechanisms remain unknown.

Myeloid cells are well known play a major role in ischemic AKI (review Han SJ, Kidney Res Clin Pract, 2019; Singbartl K, Seminars in Nephrology, 2019). Therefore, we decided to focus on macrophage/DC biology and did not follow up on B cell-related mechanisms. We have now added functional data about microvascular phagocyte subsets including M2 polarization, phagocytosis and TNF secretion (new main figures 6 and 7, and our response below). Our data highlight a new role of microvascular immune cells in maladaptive repair after AKI. These findings could provide new insights into the clinical observation that AKI is a risk factor for recurrent AKI that affects up to 30% of AKI patients.

We added some of the reviewer's remarks about B cells to the discussion.

Additional points:

1. The most notable discrepancy between cells in the peripheral blood and microvasculature in Figure 4d was the side scatter (SSC) high monocytes that the investigators describe only as being "more granular and represent a macrophage-like phenotype". No additional stains or data are present to support this designation.

We conducted new experiments to further delineate the cell subsets of the microvascular SSC^{hi} cluster using F480, CD11c and CD11b as common surface markers (gating strategy is now shown in supplemental figure 2c). CD11b⁺ NK cells were excluded in this analysis (lineage negative gating). The data show that the SSC^{hi} leukocyte include a heterogeneous mix of myeloid cells, many of which show a monocyte-like (CD11b^{hi} F480^o), macrophage (F480^{hi} CD11c^{+/-}) or dendritic cell signature.

This result is validated by the single cell sequencing data shown in figure 2 that delineate different subtypes of mononuclear phagocytes (monocytes, macrophages, DCs). We added these data to figure 1 and discuss the conclusions in the manuscript.

2. The designation of monocytes in the flow cytometry data is concerning. These cells are depicted as being Lin- (what antigens make up Lin- for this condition is not made clear in the main figure, legend, supplement, legend or the methods section), CD11b+ and Ly6Clo. The data in Figure 1D very nicely shows, with in vivo and ex vivo CD45 staining, that some cells in the tissue project out into the lumen of the microvasculature. This could be due to tissue resident dendritic cells many of which express CD11b (Mesnil, et al. 2012). The presence of these cells is supported by the scRNAseq data that clearly define cDC2 populations. Without including CD11c in the flow cytometry panel, DCs cannot be excluded from monocytes.

We apologize for the missing lineage marker information (Lin-) that is now added to the method section. For studying myeloid cells, we used NK1.1, CD20 and TCRb as negative lineage markers in the flow cytometry analysis.

Blood borne naive monocytes also express CD11c but typically in low numbers (Drutman SB, JI, 2012 figure 1; Swirski FK, JCI 2007; Tacke F, JCI 2007). CD11c is also expressed in a subset of renal F480+ macrophages (Cao Q, JASN, 2015). These limitations need to be considered when interpreting CD11c gating in monocytes.

Our flow cytometry data show microvascular CD11c medium cells as SSC low, whereas all CD11c high cells are also SSC high (figure below). Further gating reveals that SSC high CD11c high cells (purple) are also MHC-II and F480 high, pointing to a macrophage-like phenotype. In contrast, SSC low CD11c medium cells (turquoise) are F480 low and MHC-II high, resembling a typical monocyte-like phenotype. The subset gating shown above (point 1) also provides further details about the SSC high subset. However, we agree with the reviewer that the flow cytometry is not fully conclusive, and rephrased the conclusions about the SSChi population in the result section. A higher resolution is provided by the scRNAseq data in figure 2 that readily discerns the monocyte and DC/macrophage populations.

3. The methodology and use of i.v. CD45 antibodies for labelled is aligned with the field, and is common practice in studies focused on peripheral vs tissue resident immune cells. Though there is a pitfall in this methodology. As the authors note (Yatim, et al. 2016; Vollmann, et al. 2021), there are cells that extend protrusions out into the microvascular lumen, these cells would be labeled in vivo by i.v. CD45 antibodies, as such these cells will provide a false positive in the cells designated as cells of the microvasculature (Figure 1d) skewing the data and any possible interpretations. This caveat should be discussed.

We agree that the experimental approach cannot distinguish between intraluminal protrusions from perivascular cells, marginated cells and fully intravascular cells. However, we do not consider these phenotypes "false positives". We used this approach because we were primarily interested in the question of how circulating factors could trigger organ injury. Kidney injury is a known clinical complication in many inflammatory diseases such as sepsis, acute liver failure, heart failure and lupus erythematoses (among others). In these cases, there are immune complexes, pathological antibodies or DAMPs that can trigger kidney injury but mechanisms are largely elusive. Our study now suggests a new mechanism that entails direct interaction of circulating factors with organ-specific microvascular leukocytes. Whether this involves intraluminal protrusions (as shown in figure 1d) or marginated cells needs to be studied in context of the disease model. We hope that our findings help the research community to find new microvascular mechanisms of organ inflammation that can be therapeutically exploited. These thoughts are outlined in the discussion.

We agree with the reviewer that the wording "microvascular leukocyte" is somewhat misleading if it is also meant to represent perivascular cells with luminal extensions. We rephrased some conclusions in the manuscript accordingly.

4. Although data are presented for splenic microvasculature, the concept of microvasculature and the technique of pulse labeling being confined to the vasculature is questionable for spleen with about 90 percent of the blood flowing through an open sinusoidal route of circulation.

We thank the reviewer for pointing out this important exception of the spleen. We emphasized this point in the manuscript.

5. There is a disconnect between the text references to supplemental tables. The text refers to supplemental table 1 as listing the 99 custom additional added genes and then on the same page as listing 27 CD45+ leukocyte clusters. No other tables are referenced. Yet there is a supplemental table with 26 leukocyte clusters and another with 29 leukocyte clusters.

We thank the reviewer for catching this error. We fixed the references. Supplemental table 1 now lists the individualized gene panel and reagents. Supplemental table 2 shows the cell distributions in the scRNAseq data sets. Supplemental table 3 is new and shows all differentially expressed genes of all clusters in all data sets.

Overall, this is a highly observational set of experiments. The data are presented unsystematically and it is difficult to piece together continuous paths of given cell types. For example, the large population of potentially novel endovascular B cells in healthy kidneys is not discussed relative to the models of inflammation. Insights about any possible functional significance of the novel finding of endovascular B cells and their responses to AKI would be valuable especially in the light of recent evidence that CCL7 producing B cells mediate ischemia-reperfusion injury in the lung (Farahnak, et al. 2024). More comprehensive characterization of the endovascular B cells might allow the investigators to use depleting antibodies or other approaches to intervene in models of AKI and provide evidence of causality.

As outlined above, our revisional experiments focused on macrophage/DC biology because these cells are well known to regulate injury and repair in ischemic AKI (Maryam B, *Kidney360*, 2024; Kurts C, *Nat Rev Nephrology*, 2020). We now show new data about microvascular phagocyte functions of the kidney (new main figures 6 and 7). In these experiments, using the intravascular labeling approach, we can directly compare tissue resident and microvascular macrophages and dendritic cells. We observed a breakdown of the microvascular interface after AKI. The large population of homeostatic C1q+ macrophages retract and accumulate in the tissue, while up to 4 new macrophage/DC populations infiltrate the microvascular interface after AKI (figure 6). Microvascular (but not tissue-resident) F480+CD11c+ macrophages become M2 polarized and increase their capacity to phagocytose blood borne bacteria (iv injected pHrodo E.coli bioparticles). In addition, TNF secretion (baseline and in response to bacterial LPS) is increased after AKI particularly in microvascular CD11c+F480- dendritic cells while being absent in tissue resident counterparts.

DCs have been identified as main TNF producing cells (Dong X et al, *Kidney Int* 2007). TNF is known to trigger neutrophil influx and renal tubular epithelial cell apoptosis in ischemic AKI further exacerbating kidney injury (Donnahoo KK et al, *Am J Physiol* 1999; Misseri R et al. *Am J Physiol Renal Physiol.*, 2005). Our data suggest that TNF-producing DC predominantly occupy the microvascular niche after AKI regeneration. Therefore, we interpret our findings as microvascular maladaptions after injury. These findings could provide new insights into the clinical observation that AKI is a risk factor for recurrent AKI that affects up to 30% of AKI patients.

A graphical summary is now added in figure 7:

The missing “continuous path of given cell types” results from our focus on the concept of microvascular immunity rather than a specific cell type. We analyzed the endovascular B cell population somewhat more in detail, because it was the only population that was exclusively upregulated in the renal microvasculature (i.e. absent in the blood and tissue). While the evidence of CCL7 producing B cells in IRI-AKI is intriguing (Farahnak, et al. 2024), we were not primarily interested in B cell biology and feel that the experimental investigation of endovascular B cells in kidney homeostasis and injury is beyond the scope of this manuscript.

Reviewer #3 (Remarks to the Author):

General Comments: This manuscript addresses a very interesting question regarding what we are measuring when we analyze leukocyte populations isolated from various organs. The premise of the paper is that cells in the microcirculation are not equivalent to those in the peripheral blood, with the latter defined as blood obtained from a major vessel or from a cardiac ventricle. The methodology employed is entirely dependent on the veracity of the method used to label intravascular cells. The authors validated the technique by injected antibody specific for an epithelial marker and finding no staining in the extravascular space. The extravascular leukocytes, then, are identified by staining with CD45.2 anti-allotypic antibody and comparing with CD45 total staining. The cell types were then separated using flow cytometric sorting. If this methodology is as discriminatory as it appears to be, then the data indicate a variable in evaluation of intraorgan leukocytes that has not been previously considered in this manner. While this reviewer might disagree with some of the minor details, the general conclusions are well supported by the data and warrant further study.

We thank the reviewer for the time, thorough review of our manuscript and many suggestions to improve our work.

We now added another set of validation experiments using a different epithelial marker, E-cadherin (supplemental figure 1). We could not detect extravascular staining of iv administered E-cadherin in all organs (kidney, liver, lung) in both untreated and AKI-treated mice. The spleen is an exception because of the sinusoidal anatomy that lacks a classical microcirculation. Published work used a similar approach to tag intravascular cells (Potter EL, Science Transl. Med 2021; Prior JT, Vaccines 2023; Anderson KG, 2014). It also needs to be considered that this method cannot distinguish between perivascular cells with intraluminal protrusions, marginated cells or fully intravascular cells. However, we agree with the reviewers' assessment that the main conclusions hinge on the validity of the applied methodology. This aspect is critically discussed in the manuscript ("limitations"). We think that the overall conclusion of a complex disease- and organ-specific microvascular immune landscape is valid despite these caveats.

We also show new data about microvascular phagocyte functions of the kidney (new main figures 6 and 7). In these experiments, using the intravascular labeling approach, we can directly compare tissue resident and microvascular macrophages and dendritic cells. We observed a breakdown of the microvascular interface after AKI. The large population of homeostatic C1q+ macrophages retract and accumulate in the tissue, while up to 4 new macrophage/DC populations infiltrate the microvascular interface after AKI (figure 6). Microvascular (but not tissue-resident) F480+CD11c+ macrophages become M2 polarized and increase their capacity to phagocytose blood borne bacteria (iv injected pHrodo E.coli bioparticles). In addition, TNF secretion (baseline and in response to bacterial LPS) is increased after AKI particularly in microvascular CD11c+F480- dendritic cells while being absent in tissue resident counterparts. These data emphasize that microvascular and tissue-resident leukocytes can have different functions in AKI.

We have now added a graphical abstract to summarize our findings (figure 7):

Specific comments:

1. Using a gradient for isolation of cells can significantly skew the results. The authors avoided this common pitfall.
2. Paragraph starting with: “To detect unsupervised phenotypes and transcriptional states of microvascular leukocytes...” How the data were multiplexed and processed could be clarified. It seems the data are derived from a total of four mice and the data from each mouse was multiplexed with renal tissue leukocytes, microcirculation and peripheral blood? For tissue, then, about 1500 leukocytes were sequenced per kidney. Were the contributions from each mouse roughly equal?

The cell numbers per mouse are shown below. Because the cell compartments were not equal in the untreated condition (roughly 2:2:1 for blood, microvascular and tissue leukocytes) we also added another control using data downsampling to 1:1:1. Here, our scRNAseq results in figure 2 could be confirmed. These information were added to the manuscript (methods and results section).

	untreated				AKI-reg			
	M1	M2	M3	M5	M4	M6	M7	M8
Blood	1453	2084	5041	5249	5533	4730	4805	4922
MV	2191	3481	4801	2935	2870	2969	3495	4102
Tissue	571	895	2654	2118	2473	2122	4795	3039

3. It is stated that Adgre1+Itgax+Fcgr4+ macrophages in the kidney were almost entirely microvascular. The data in figure 2e (cluster 0) do not support that interpretation because the difference (in proportion) are not statistically significant. In considering proportion, the figure does not state what the denominator is – CD45+ total? Also, if one is to make such a statement, then the data to be compared should be absolute numbers.

We thank the reviewer for pointing this out. Most microvascular cell types were also detectable in the tissue (i.e. no significant difference between microvasculature and tissue), and this is now clearly stated in the manuscript. The significance for each cluster and data set is shown in supplemental figure 4, 7 and 8, and figure 3.

We found that 67% of all renal macrophages and DC are in contact with the microcirculation (figure 6d).

The Y axis in figure 2e is the leukocyte fraction (of all CD45+ cells). We did not use absolute numbers as the input cell numbers for the groups MV, tissue and blood differed. We rephrased the conclusions in the manuscript accordingly.

4. For the pseudotime measurement, it would be helpful to the reader to specify how the initial point was chosen.

Cluster 0 was chosen as a starting point because it is the largest blood borne cell cluster that is significantly enriched in the blood compared to the microvasculature. This information was added in the text and the significance is now visualized figure 3d. We then used all significantly up- or downregulated clusters (1,4,6,7 versus 0,5) for the functional annotation analysis in figure 3f.

5. Paragraph: “Myeloid cells are major effectors in ischemic AKI”. It is stated “phagocytes were absent in the peripheral blood...”. Phagocytes is a vague term in this context and essentially encompasses almost all myeloid cells. Is that the intended meaning?

We deleted the term phagocyte and replaced it with “macrophages and dendritic cells”.

Point to point response

Revision of the manuscript NCOMMS-24-03267A

" Microvascular immunity is organ-specific and remodeled after kidney injury"

Reviewer #1

The authors have addressed almost all of my concerns satisfactorily. Only the first point needs some more editing: They responded to it:

"We believe that this is a misunderstanding. Two cell populations (or groups) were separated from the kidney: microvascular (CD45+CD45.2iv+) and tissue (CD45+CD45.2iv-) leukocytes. The former group was then compared to a third group, namely peripheral blood, that was separately collected from the right heart (figure below)."

The authors did not get my point: I pointed out that their microvascular cells must contain some vascular cells, ie cells in large vessels located within the kidney. These are not in the microcirculation, but in essence equivalent to the cells in their third group, the blood. Fig 2c shows quite clearly overlap between several clusters of blood and their microvascular cells, especially in monocytes, B and T cells, ie the cells expected to be present within vessels. Thus, not all their microvascular cells are microvascular cells; some are in fact blood cells. Those that differ in scRNASeq from blood cells will of course show differences from blood cells, by definition. They have removed this circular reasoning from their paper and focussed their manuscript on cells present only in the microcirculation. I am satisfied with this. But I would recommend to more clearly mention that this contamination exists and discuss it.

We agree with the reviewer. As suggested, we emphasize this issue in the manuscript (methods section, line 58).

Reviewer #2

The investigators have extensively revised the text, performed additional experiments and added multiple new figures including a graphical summary in response to the reviews. Specifically, they have added a bar graph depicting an ontological analysis of microvascular B cells (figure 3f). The investigators forego further analysis of B cells because "tools to target intravascular B cells specifically are still lacking, and therefore, the precise functions and mechanisms remain unknown." Instead, they have chosen to focus on macrophages and DCs. To this end, they have further delineated the cell subsets of the microvascular SSChi cluster using F4/80, CD11c and CD11b, and represented the data in a stacked bar graph (figure 1e). They also reanalyzed the macrophage and DC RNAseq data (figure 6) as well as performed a functional assay of E Coli phagocytosis (Figure 7).

While these changes provide some added information, the paper primarily presents a catalogue of data and misses opportunities to make a cohesive story.

For example, in figure 6 the subclustering of macrophages and DCs is presented merely by multiple different graphics illustrating increase or decrease relative to perturbation and location. The authors present data in figure 6c, e and f indicating that cluster 2 is increased 12 days after AKI in the microvasculature. Instead of using their own data to determine what genes might be differentially expressed in the cluster in response to AKI, they perform flow cytometry using "the common markers CD11c and F4/80" in figure 6h. The CD11c+F4/80+ subpopulation in flow is somewhat counterintuitive given that cluster 2 has low expression of Adgre1 (F4/80) and high expression of Itgax (CD11c), whereas clusters 0,1 and 6 have high expression of Adgre1 (F4/80) and are low expressors of Itgax (CD11c). Similarly, the identification of the phagocytic cells in figure 7 could be better integrated with the clusters defined in figure 6 with a more judicious choice of cell markers.

The increase of TNF-secreting cluster 2 macrophages as the predominant microvascular phagocyte subtype after kidney injury is a main finding of our experiments. Figure 6a-f present main characteristics of these cells including differentially expressed genes (fig. 6a, supplemental table 3). The top 10 DE genes include: Kcne3, F13a1, Cd163, Clec10a, Thbd, Cd244, Ifitm3, Nupr1, Cd300a, Crip1. We tested CD163, CD300, Clec10a, among others, but could not yet identify a unique surface marker or marker combination that exclusively labels cluster 2 cells. This is a common issue in high resolution RNA-seq experiments since RNA expression only variably correlates with protein expression (Schwanhausser B et al, Nature 2011; Liu Y et al, Cell 2016).

F480 and CD11c are widely used as surface markers for macrophage/DC. Using these markers is helpful to compare our findings with published data. Thus, we analyzed the Adgre1 (F480) and Itgax (CD11c) gene expression in all clusters as shown in the heatmap (figure 6b, and figure below). We would like to point out that "yellow/beige" marks a medium expression and blue codes for low expression. Accordingly, cluster 2 shows a moderate and high expression intensity of Adgre1 (F480) and Itgax (CD11c), respectively. For reference, cluster 3 shows a negative Adgre1 expression (blue). A double positive signature is present in cluster 0, 1, 2, 4 and 5. Cluster 3 and 6 are labeled by F480-CD11c+ and F480+CD11c-, respectively.

Review figure: Relative Adgre1 (F480) and Itgax (Cd11c) RNA gene expression in microvascular macrophage/DC cell cluster subsets. Related to main figure 6a.

Immunofluorescence imaging with intravascular antibody labeling validates that the CD11c/F480 staining strategy readily discerns three different microvascular cell populations including the double positive group, F480-CD11c+ and F480+CD11c- cells (figure below). Flow cytometry confirms that about 57% of microvascular CD11c+ cells in the kidney also express F480 (figure below). Since cluster 2 becomes the most abundant cell type in AKI-reg (figure 6d) and is characterized by Adgre1 and Itgax gene expression, we used the double positive staining (F480+CD11c+) for identification.

a untreated,
microvascular antibody staining

b untreated,
renal microvascular CD45+ cells

Review figure: Intravascular F480 and CD11c staining to dissect macrophages/DC cell types in the renal microvasculature. a) Microvascular staining of F480 and CD11c shows a heterogeneity of F480+CD11c- (red), F480+CD11c+ (yellow) and F480-CD11c+ (green) macrophages/DCs. b) About 15% of all renal microvascular leukocytes are CD11c+ positive, of which 57% are also F480+. Representative flow cytometry data using a healthy mouse kidney.

Our main goal was to identify unique functions of microvascular immune cells compared to tissue resident counterparts. Although the F480/CD11c staining only partially resolves the heterogeneity of microvascular macrophages/DCs, it suffices to identify main macrophages/DC lineages in the kidney microvasculature. We can show that these cell types exert different functions such as phagocytosis of blood borne E.coli and TNF secretion, highlighting the significance of microvascular immune responses.

We changed the manuscript to emphasize these points (result section line 389, discussion line 480).

In presenting the data in figure 6, the authors attribute function based on a selected few genes, such as cluster 0 “showed a fibrosis-related gene signature with upregulation of Col14a1 and Tgfb3 (figure 6b).” This conclusion would be better supported with a gene ontology analysis.

We thank the reviewer for this suggestion. We conducted a gene set enrichment analysis (GSEA) using significantly upregulated DE genes of cluster 0. Here, fibrosis-related pathways were not significantly enriched. However, it needs to be considered that GSEA relies on curated databases which often do not include very recent data. In fact, Tgfb3 and Col14a1 have been recently described in kidney fibrosis (Cao S et al, Nat commun 11/2023).

We deleted our more general statement and now mention the genes with their specific references (discussion, line 457).

The authors also interpret findings for perivascular C1q+ macrophages as indicating a retraction from the endoluminal space (figure 6e,f). The word retraction or retract is used several times in this context and it is not clear whether the authors mean that these cells retract intraluminal extensions, which could be documented by histological means as in figure 1d.

C1q+ macrophages lose contact to the blood stream after kidney injury. RNAseq experiments demonstrate that these cells do not disappear but relocate to the tissue compartment. The word "retract" might be misleading as the main mechanism is not a loss of intraluminal extensions. Most of CD45.2iv+ macrophages/DC in the kidney are positioned entirely in the vessel lumen (see figure below). Only a small fraction of cells features intraluminal protrusions that extend from a perivascular cell body as shown in figure 1d. Thus, we changed the word "retract" to "withdraw" and added these considerations to the manuscript (discussion, line 458).

Review figure: Most macrophages/DCs in the microvasculature are positioned entirely in the vessel lumen. a) Overview and 4 close up views of intravascular F480+ cells. F480 was stained intra- (red, fluorophore AF647) and extravascularly (green, fluorophore PE). Green = extravascular F480+ cells, yellow = double positive staining showing intravascular cells. b) Confocal imaging of 5 individual microvascular leukocytes showing that most of the cell body is located intravascularly (CD45.2 iv red, fluorophore AF647; CD45 ex vivo staining green, fluorophore PE). Representative pictures from two healthy mouse kidneys.

The graphical summary in figure 7 is helpful. However, the cluster nomenclature can lead to some confusion. For example, cluster 0 in the left panel is not the same as in the right panel. Possibly the cluster designations could be omitted.

We removed the cluster designations in the graphical summary.

Finally, the new title that indicates the findings demonstrate the changes microvascular cell content "maladapts after kidney injury" is not supported by any functional data.

TNF is a main cytokine driving kidney injury (Xu C, *Kidney Int*, 2014; Singbartl K, *Seminars in Nephrology* 2019; Al-Lamki RS, *Kidney Int* 2015; Li J, *Kidney Int* 2023). Macrophages and DCs are known TNF-producing cells (Serbina MV, *Immunity* 2003; Han HI, *Pediatr Nephrol* 2020). Ischemia-reperfusion injury of the mouse kidney triggers inflammation with a high concentration of TNF (Dong X, *Kidney Int* 2007). Here, F480+ CD11c+ macrophage/DCs were the main TNF-producing cell type, and in vivo depletion of DCs attenuated TNF secretion and tissue injury (Dong X, *Kidney Int* 2007). Moreover, kidney-protective tolerogenic DCs are known to produce less TNF (Li J, *Kidney Int* 2023).

Our data indicate that kidney injury mediates a restructuring of the microvascular interface with new TNF-producing macrophage/DC cell types. Since elevated TNF secretion is a hallmark for renal inflammation and tissue injury as evidenced by many studies, we describe these changes as "maladaptation". However, as the reviewer points out, this statement is not directly backed by functional data. Thus, we suggest to change the word "maladapt" to "remodeled" (title), and change the abstract and discussion accordingly.